# Stochastic parabolic growth promotes coexistence and a relaxed error threshold in RNA-like replicator populations

**Mátyás Paczkó[1,2], Eörs Szathmáry[1,3,4]\*, András Szilágyi[1]**

[1]Institute of Evolution, HUN-REN Centre for Ecological Research, Budapest, Hungary; [2]Doctoral School of Biology, Institute of Biology, ELTE Eötvös Loránd University, Budapest, Hungary; [3]Center for the Conceptual Foundations of Science, Parmenides Foundation, Pöcking, Germany; [4]Department of Plant Systematics, Ecology and Theoretical Biology, Eötvös Loránd University, Budapest, Hungary

**\*For correspondence:**
szathmary.eors@gmail.com

**Competing interest:** The authors declare that no competing interests exist.

**Abstract** The RNA world hypothesis proposes that during the early evolution of life, primordial genomes of the first self-propagating evolutionary units existed in the form of RNA-like polymers. Autonomous, non-enzymatic, and sustained replication of such information carriers presents a problem, because product formation and hybridization between template and copy strands reduces replication speed. Kinetics of growth is then parabolic with the benefit of entailing competitive coexistence, thereby maintaining diversity. Here, we test the information-maintaining ability of parabolic growth in stochastic multispecies population models under the constraints of constant total population size and chemostat conditions. We find that large population sizes and small differences in the replication rates favor the stable coexistence of the vast majority of replicator species ('genes'), while the error threshold problem is alleviated relative to exponential amplification. In addition, sequence properties (GC content) and the strength of resource competition mediated by the rate of resource inflow determine the number of coexisting variants, suggesting that fluctuations in building block availability favored repeated cycles of exploration and exploitation. Stochastic parabolic growth could thus have played a pivotal role in preserving viable sequences generated by random abiotic synthesis and providing diverse genetic raw material to the early evolution of functional ribozymes.

## eLife assessment

This study provides a **valuable** theoretical exploration of non-enzymatic sustained replication of RNA systems, in the parabolic growth regime of the evolution of putative primordial replicators. It provides **convincing** evidence that parabolic growth mitigates the error threshold catastrophe, thus demonstrating another way in which this regime contributes to the maintenance of genetic diversity. The findings shed light on relevant evolutionary regimes of primordial replicators, with potential applicability to our understanding of the origin of life.

## Introduction

Current knowledge of nucleotide chemistry as well as a large body of indirect evidence from recent organisms support the RNA world hypothesis; a concept that once RNA fulfilled the role of an evolvable primordial informational polymer and biochemical reaction catalyst at the same time (**Woese, 1965**; **Crick, 1968**; **Orgel, 1968**; **Orgel, 1992**; **Gilbert, 1986**; **Benner et al., 1989**; **Joyce, 2002**; **Lee et al., 2023**). However, the initial weaknesses of the original RNA world hypothesis, such as the lack of

**eLife digest** All living things use molecules known as nucleic acids to store instructions on how to grow and maintain themselves and pass these instructions down to the next generation. However, it remains unclear how these systems may have evolved from simple molecules in the environment when life began over 3.6 billion years ago.

One idea proposes that, before the first cells evolved, abiotic chemical processes gave rise to substantial building blocks of ribonucleic acids (or RNAs, for short). Over time, RNAs could have combined to form polymers of random sequences that started to copy themselves to make simple machines, only carrying the information required to make more of the same RNAs. Later on, these RNA molecules teamed up with proteins, fats and other molecules to make the first cells.

When RNA replicates, the parent molecule is used as a template to assemble a new copy. While the new RNA molecule remains attached to its template it prevents the template being used to make more RNA. Therefore, it is thought that the speed at which a specific RNA machine copied itself may have varied in a pattern known as parabolic growth. Furthermore, when RNA replicates without the help of other biological molecules, the process is very prone to errors, which would have severely limited how much information the RNA machines were able to pass on to the next generation.

Theoretical work suggested that under certain conditions, parabolic growth may favor the maintenance of a large amount of RNA sequence-coded information, but it is not clear if this is actually possible in nature. To address this question, Paczkó et al. developed mathematical models to investigate the effect of parabolic growth on the ability of RNA to replicate without other biological molecules. The models show that when large numbers of RNAs are present, small differences in how quickly different RNAs replicated favored the stable coexistence of different RNA sequences. Parabolic growth decreased the adverse effect of copying errors, allowing larger pieces of RNA to faithfully replicate themselves.

This work suggests that parabolic growth may help to maintain different types of RNA (or similar replicating molecules) in a population and in turn, help new simple life forms to evolve. In the future, these findings may be used as a framework for laboratory experiments to better understand how early life forms may have evolved.

a reliable replication mechanism and the consequential loss of heritable information (*Bernhardt, 2012*; *Szostak, 2012*; *Le Vay and Mutschler, 2019*), have prompted scientists studying the origin of life to devise a wide variety of physicochemically refined models of the RNA world. A potentially promising suggestion is that prior to template replication of complex polymers catalyzed by an RNA polymerase ribozyme (i.e. replicase), genetic information and catalytic functions were initially distributed among short sequence modules that could occasionally be ligated to increase molecular complexity in a stepwise manner (*Vlassov et al., 2005*; *Manrubia and Briones, 2007*; *Briones et al., 2009*). Replication of these short sequence modules (perhaps with some catalytic properties) could have occurred in the absence of a replicase ribozyme, driven by some template-directed, non-enzymatic replication mechanism (*von Kiedrowski, 1986*; *Zielinski and Orgel, 1987*; *Patzke and Kiedrowski, 2007*; *Leu et al., 2011*; *Leu et al., 2013*). Thus, templated, enzyme-free replication in pools of nucleic acid oligomers that resulted from random abiotic polymerization may have contributed to the emergence of structured oligonucleotides during prebiotic evolution (*Le Vay and Mutschler, 2019*; *Rosenberger et al., 2021*). For example, it has been experimentally demonstrated that activated oligonucleotides can act as catalysts for non-enzymatic replication of RNA containing all four nucleotides, with the fidelity sufficient to sustain a genome size large enough to encode active ribozymes (*Prywes et al., 2016*).

If enzyme-free replication of oligomers with a high degree of sequence variability was indeed attainable from prebiotic chemistry, then it becomes a vital issue how a critical level of diversity of the associated genetic information (*Szostak, 2011*) could be preserved under competition among distinct replicator types with different competitive abilities (e.g. replicabilities) (*Eigen, 1971*; *Szathmáry, 1991*; *Maynard Smith and Szathmáry, 1995*; *Jiménez et al., 2013*; *Ruiz-Mirazo et al., 2017*). This recognition calls for elucidation of the coexistence mechanisms that could have alleviated the competition among independently replicating information carrying modules. For instance, parabolic growth dynamics, a kinetic behavior observed in non-enzymatic self-replicating systems (***von***

*Kiedrowski, 1986*), has been proposed as an ideal candidate mechanism to sustain a large amount of prebiotic genetic information (*Szathmáry and Gladkih, 1989*; *Scheuring and Szathmáry, 2001*). In this kinetics, the growth order $p$ is equal or close to 0.5 (i.e. the dynamics is sub-exponential) because increased stability of the template-copy complex (rate of association divided by dissociation) promotes parabolic growth (*von Kiedrowski et al., 1991*; *von Kiedrowski and Szathmáry, 2001*). This is in sharp contrast to exponential growth, where $p = 1$ implies maximum dissociation that allows for fast autocatalysis (*Tjivikua et al., 1990*; *von Kiedrowski et al., 1991*). Thus, replicating individuals in populations displaying parabolic growth kinetics are inherently prone to self-inhibition by duplex formation, a feature that can efficiently damp competition and therefore promote coexistence. Indeed, parabolic population growth can sustain an unlimited number of competing replicator species in the infinite population size limit (*Szathmáry and Gladkih, 1989*; *Varga and Szathmáry, 1997*). However, molecular evolution of the Darwinian type is thought to necessitate exponential, rather than parabolic amplification, because in the latter case robust coexistence of the competing replicator species generally limits the efficiency of natural selection (*Sievers and von Kiedrowski, 1998*; *Szilágyi et al., 2017*; *Strazewski, 2019*). Therefore, a major current focus in models of parabolic replication is how to incorporate Darwinian selection into its dynamics (*Lifson and Lifson, 1999*; *Scheuring and Szathmáry, 2001*; *von Kiedrowski and Szathmáry, 2001*).

While the latter models considerably improved the parabolic replication model framework in terms of its potential to incorporate evolvability, they did not account for the copying error threshold of their respective systems; i.e., an error rate of the replication mechanism at which the system's ability to stably propagate genetic information from generation to generation is prohibited, resulting in irreversible loss of sequence-coded information (*Eigen, 1971*; *Joyce, 2002*; *Szostak, 2012*). The possibility that high mutation rates could lead to such an 'error catastrophe' is one of the major caveats surrounding the RNA world hypothesis, given that the first RNA replication mechanisms have probably been inherently error-prone (*Eigen, 1971*; *Manrubia and Briones, 2007*; *Leu et al., 2011*).

Moreover, practically no study is known to have treated parabolic replication so that physicochemical details and ecological constraints, together with stochastic effects (that supposedly also have considerable impact on finite systems), would have been taken into consideration. The present study aims to remedy these deficiencies by quantifying the heritable information-maintaining potential of finite populations of unlinked, RNA-like template replicators displaying parabolic growth modeled

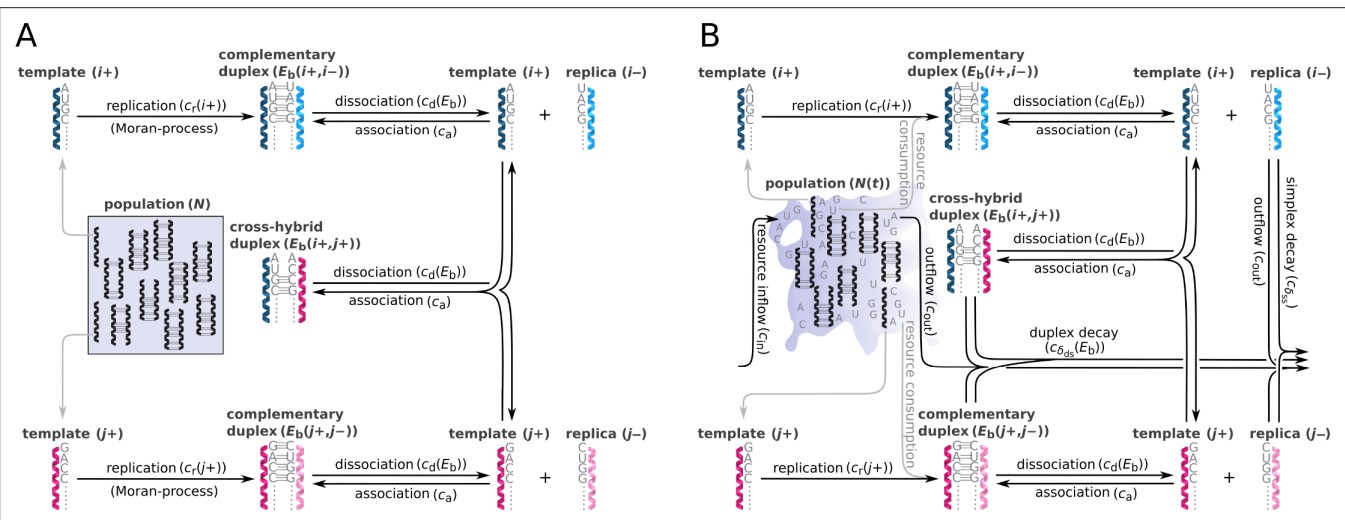

**Figure 1.** Schematic representation of two parabolically replicating systems. (**A, B**) These systems represent spatially constrained or aggregated, but well-mixed (e.g. by wet-dry cycles on mineral surfaces) replicator populations. Black arrows indicate the possible chemical reactions, $i$ and $j$ denote master types. (**A**) Constant population model representing a Moran process. $N$ denotes the (fixed) total number of replicator molecules. This model allows us to investigate the inherent dynamical properties of parabolic growth, such as its diversity-maintaining ability in face of mutations, under fixed and controlled conditions. (**B**) Chemostat model representing an open system. $N(t)$ denotes the time-dependent total number of replicators. This model allows us to investigate the effects of the changes in prebiotic environments on parabolic coexistence. Such alternations in environmental conditions could have involved, for example, variations in resource availability and the consequential shifts in the strength of resource competition. For the explanation of mechanisms and constants, see main text and *Table 1*.

as stochastic dynamics. We represent the replicator sequences individually, and copying error as well as energetic constraints in strand separation are also considered explicitly. Using a constant population model version of this framework, we first investigate how the diversity-maintaining mechanisms operate in the parabolic regime in a finite population of replicators. We further demonstrate that parabolic coexistence is resistant to mutation rates assumed for template-directed, enzyme-free replication, thereby proposing a simple biochemical mechanism to relax the error threshold in hypothetical information storage systems of the RNA world. In order for our stochastic simulation results to be comparable with the theoretical error threshold of parabolic dynamics, we analytically calculate the equilibrium master (i.e. fittest sequence) concentrations as a function of the copying fidelity and the reproductive superiority of the master sequences relative to the mutants. We also compare the error threshold in case of parabolic dynamics to the case of the exponential dynamics. Finally, in order to validate that Darwinian selection by competitive exclusion is also feasible in this framework, we consider a chemostat model of parabolic replication, in which monomer building block resources are additionally taken into account. Using this approach, we demonstrate that increased competition, mediated by a decrease in the resource influx rate, can induce a switch from the coexistence ensured by parabolic dynamics to survival of the fittest.

## Methods
### Parabolic replicator model framework

We consider populations of oligomer molecules based on nucleotide-like monomer constituents (to preserve generality for potential informational polymers other than RNA). Individual molecules are represented by their primary (monomeric sequence) structures and are assumed to undergo template-directed, non-enzymatic replication. This replication mode converts a single-stranded plus- or minus-strand template into a template-copy duplex. Strand separation rates of these duplexes are relatively low compared to strand association (e.g. *von Kiedrowski et al., 1991*). Therefore, replication cycles are strongly inhibited and as a result, the population dynamics displays parabolic characteristics. We define a set of replicators with the highest fitness relative to all mutant genotypes and refer to them as 'master' types (*Eigen, 1971*), representing the genetic information to be maintained. To analyze the diversity-maintaining ability of parabolic dynamics in a competitive setting, the master replicator types are characterized by different replicabilities. This assumption reflects the fact that, at least in case of RNA, the rate of copying of short templates by polymerization of activated monomers inherently varies with template composition (*Szostak, 2012*). In order to make our model framework to be more suitable for focusing on different aspects of parabolic dynamics, we considered two different model versions of parabolic replication (*Figure 1A and B*).

### Constant population model of parabolic replication

In the constant population model (*Figure 1A*), there is no exchange of materials with the environment (monomer resources are not represented explicitly) and a constant total population size is ensured by implementing a Moran process into the replication mechanism (*Moran, 1958*). Accordingly, each newly synthesized strand replaces a sequence randomly chosen from the population (that can be either in simplex or duplex state; in the latter case the population size decreases by one until the next replication). The dynamics is based on the following three reaction categories: replication of a single strand $S_i$ (*Equation 1A*); association of a single strand $S_i$ with another strand $S_j$ (*Equation 2A*), and dissociation of a double strand $S_iS_j$ (*Equation 3A*). The rate constants corresponding to these three reactions are $c_r$, $c_a$, $c_d$, respectively. Mutation events are taken into account during replication. These simple assumptions are in line with the kinetics of template-directed enzyme-free nucleic acid replication (for an experimental system, see, for example, *Sievers and von Kiedrowski, 1998*). The $n \cdot R$ expression in *Equation 1A* and *Equation 1B* denotes the amount of monomer building blocks needed for replication.

$$S_i \left( +n \cdot R \right) \xrightarrow{c_r} S_iS_j \tag{1A}$$

$$S_i + S_j \xrightarrow{c_a} S_iS_j \tag{2A}$$

$$S_iS_j \xrightarrow{c_d} S_i + S_j \tag{3A}$$

## Chemostat model of parabolic replication

In case of the chemostat model (**Figure 1B**), a constant inflow of monomer building blocks is assumed to supply the synthesis of the new strands, i.e., the dynamics describes an open system (**Schuster, 2011**). This inflow component, together with type-specific replicability rate constants and a decay component on the replicators as well as outflow of the excess production, determines the time-dependent total number of replicators during the dynamics. The three basic reaction categories are the same as in the constant population model, namely: replication (**Equation 1B**), association (**Equation 2B**), and dissociation (**Equation 3B**), however, the dynamics of the resources is explicitly represented. The common inflow rate of all the four RNA nucleobases is $c_{in}$ (**Equation 4B**). In contrast to the constant population model, here we assume decay of both single and double strands, the decay rates are denoted by $c_{\delta_{ss}}$ and $c_{\delta_{ds}}$, respectively, see **Equation 5B** and **Equation 6B**. The type-independent outflow of molecular species is characterized by $c_{out}$ (**Equation 7B**).

$$S_i + n \cdot R \xrightarrow{c_r} S_i S_j \tag{1B}$$

$$S_i + S_j \xrightarrow{c_a} S_i S_j \tag{2B}$$

$$S_i S_j \xrightarrow{c_d} S_i + S_j \tag{3B}$$

$$\xrightarrow{c_{in}} R_{(A, U, G, C)} \tag{4B}$$

$$S_i \xrightarrow{c_{\delta_{ss}}} \varnothing \tag{5B}$$

$$S_i S_j \xrightarrow{c_{\delta_{ds}}} \varnothing \tag{6B}$$

$$S_i S_j,\, S_i,\, R_{(A,U,G,C)} \xrightarrow{c_{out}} \varnothing \tag{7B}$$

In both model versions, a population consists of replicators with equal length $L = 10$, composed of A, U, G, C nucleobases. We consider randomly generated master sequence sets; however, we impose

**Table 1.** Model parameters.

| Parameter | Definition | Values or range (default) |
|---|---|---|
| $N$* | Total replicator population size | $\{10^3;\ 10^4;\ 10^5\}$ |
| $\delta$* | Replicability distance | $\{0.001;\ 0.005;\ 0.01\}$ |
| $p_{mut}$* | Per base mutation rate | $\{0.01;\ 0.05;\ 0.1;\ 0.15\}$ |
| $f$*† | Duplex decay factor | $\{0.01;\ 0.1;\ 0.5;\ 1\}$ |
| $c_{in}$*† | Kinetic constant for resource influx | $\{0.01;\ 0.1;\ 1\}$ |
| $c_{\delta_{ss}}$† | Kinetic constant for simplex decay reactions | $10^{-4}$ |
| $c_{\delta_{ds}}$† | Kinetic constant for duplex decay reactions | $[1.15 \cdot 10^{-8},\ 6.4 \cdot 10^{-5}]$ |
| $c_{\delta_{out}}$† | Kinetic constant for outflux of molecular species | $[1.75 \cdot 10^{-6},\ 10^{-5}]$ |
| $c_r$ | Kinetic constant for replication reactions | $[0.005,\ 0.095]$ |
| $c_a$ | Kinetic constant for association reactions | 1 |
| $c_d$ | Kinetic constant for dissociation reactions | $[0.01,\ 0.6309]$ |
| $T$ | Number of master replicator types | 10 |
| $L$ | Sequence length | 10 |
| $\varepsilon$ | Error factor for the Hamming distance $(\Delta_{ij}) \in \{2;1;0\}$ mutant classes | $\{0.05;\ 0.2;\ 1\}$ |
| $\phi_{min}$ | Minimum master replicability | 0.005 |
| $\kappa$ | Scale factor for duplex dissociation probability distribution | 0.15 |

*Model parameters involved in screening.

†Parameter used only in the chemostat system model.

the constraint that the Hamming distance between any two master sequences must be at least equal to two, to avoid source-sink-type dynamics.

We also generated sequences containing a predefined amount of G+C nucleobases (Figure 3A and B), which would lead to increased probability of obtaining compositionally similar random sequences (i.e. generally smaller pairwise Hamming distances between masters, and between masters and their complementary sequences). In order to circumvent this distortion, the nucleobases were randomly shuffled among the loci until the Hamming distance >3 condition was satisfied for each distinct master-master and master-complementary sequence pair and for the 10 master-complementary sequence pairs of the same type, thereby excluding palindromes.

## Replication and mutation

To analyze the diversity-maintaining ability of the system, we generate a $T$-membered set of master sequences. The $\phi_i$ replicability of master sequence type $i$ is:

$$\phi_i = \phi_{\min} + (i - 1) \cdot \delta \tag{8}$$

where $i = 1, 2, \ldots, T$. Note that the difference between the lowest and highest replicability is $(T - 1) \cdot \delta$, thus, $\delta$ can be interpreted as *replicability distance*. Mutations reduce the replicability of master sequences by a predefined error factor: $\phi = \varepsilon \cdot \phi_i$, where in case of one mutation, $\varepsilon = 0.2$, and with two mutations, $\varepsilon = 0.05$. A $j$ sequence with more than two Hamming distance to *any* $i$ master sequence ($\Delta_{ij} > 2$) has a baseline replicability: $\phi_j = 0.05\phi_{\min}$. To determine the replicability of a given replicator, we compute its replicability against all master types and choose the highest value (for instance, if a replicator is a one-error copy of master type 5, a two-error copy of master type 3 and has more than two errors to all other master types, the replicability of the replicator is $0.2\phi_5$). This rule is implemented into the model to treat the (very rarely occurring) situation, when the $\Delta_{ij}$ Hamming distance of a newly synthesized $j$ strand is smaller than 3 to more than one $i$ master sequences. For simplicity we assume that the rate constant of replication (see *Equation 1A* and *Equation 1B*) is equal to the replicability: $c_r = \phi$. Substitution mutations can occur during the replication with the probability $p_{\mathrm{mut}}$ per base. Because of the fixed sequence length, deletion and insertion are not permitted.

## Dissociation, decay, and association

Binding energy ($E_b$) of a duplex (in arbitrary units) is determined by the number of Watson-Crick pairs present in the double strand according to

$$E_b = n_{AU} + 2n_{GC}, \tag{9}$$

where $n_{AU}$ and $n_{GC}$ are the numbers of A-U and G-C pairs, respectively (frameshift is not allowed). The relatively high melting temperature of GC-rich double-stranded RNAs (e.g. *Freier et al., 1986*) is considered here by including a prefactor 2 in the H-bond strength of a G-C pair, which renders the binding energy of this pairing to be the double that of an A-U pair. Note that the maximum binding energy is consequently $2L$. If $n_{AU} + n_{GC} < 2$, association is not possible due to low duplex stability. As in our model, dissociation of the replicators is an enzyme-free process, the main driving force of the strand separation mechanism is temperature. Consequently, the dissociation probability ($p_{\mathrm{diss}}$) of a duplex with binding energy $E_b$ follows the Boltzmann distribution:

$$p_{\mathrm{diss}}(E_b) = \frac{\exp(-\kappa E_b)}{\sum_{E=2}^{2L} \exp(-\kappa E)}, \tag{10}$$

where $\kappa$ incorporates the (inverse) temperature and other possible environmental factors that can have potential effects on duplex separation (e.g. ionic concentration, $p$H, etc.), see, for example, *Le Vay and Mutschler, 2019*. In order to speed up the convergence, we set the $\kappa$ value so that the dissociation probability of a duplex with maximum binding energy is 0.01 (i.e. $p_{\mathrm{diss}}(2L) = 0.01$). For sake of simplicity we assume that the rate constant of dissociation is equal to the dissociation probability: $c_d = p_{\mathrm{diss}}$, see *Equation 3A* and *Equation 3B*.

The degradation rate of single strands ($c_{\delta ss}$) is constant, regardless of the sequence. The decay rate of duplex molecules is affected by the binding energy according to

$$c_{\delta_{ds}} = f \cdot c_{\delta_{ss}} \cdot 0.8^{E_b}, \tag{11}$$

where the duplex decay factor parameter $f$ allows us to scale the duplex decay rates relative to that of the simplexes. By assuming $0 < f \leq 1$ and satisfying the condition of $E_b \geq 2$ for every duplex (which is ensured by the $n_{AU} + n_{GC} \geq 2$ condition), simplex molecules have higher decay rate than duplexes. This model assumption reflects the fact that the structure of double-stranded RNA provides a certain degree of stability against alkaline hydrolysis in solutions relative to that of single-stranded RNA (*Zhang et al., 2021*).

The association rate constant is sequence-independent: $c_a = 1$ for all pairs of single strands. Cross-hybridization between not fully complementary strands is allowed, the binding energy in this case is also computed according to *Equation 9*.

## Stochastic simulation algorithm

In order to take finite population sizes and stochastic effects into account, we consider an individual-based, stochastic approach of parabolic dynamics. Accordingly, the model is implemented as a Gillespie stochastic simulation algorithm (*Gillespie, 1976*; *Gillespie, 1977*) adapting the optimized direct method formulation (see, for example, *Cao et al., 2004*, for details). The algorithm randomly chooses a reaction to take place, based on a propensity function that is specified for every $\mu$ reaction channel as:

$$\alpha_\mu(t) = h_\mu c_\mu, \tag{12}$$

where $\mu = 1, \ldots, M$, and $M$ is the number of possible reaction channels, $c_\mu$ denotes the corresponding rate constant, $h_\mu$ is the number of the combinatorially distinct potential configurations of $\mu$ reaction-compatible molecules at time $t$. $\tau$ denotes the time when the next reaction occurs after $t$. This quantity is calculated as a function of the summed propensities:

$$\tau = \frac{1}{\sum_{\mu=1}^{M} \alpha_\mu(t)} \ln \frac{1}{r}, \tag{13}$$

where $r$ is a uniformly distributed random number from the [0, 1] interval. After one of the reactions is chosen and takes place, the frequencies of the corresponding chemical species are updated based on the reaction kinetic equations (*Equations 1A–3A* and *1B–7B*). Then the respective propensity functions are also updated and this procedure is repeated until reaching the termination condition.

In all simulations, initially each master type is present in equal copy numbers in simple-stranded form. We monitor the frequency of the master types, the replicability of all replicators including mutant copies of the masters, the number of surviving master types, the simplex-duplex asymmetry, and the total number of replicators (in the chemostat system). The steady-state characteristic of coexistence, i.e., whether the number and the identity of the surviving master types are in dynamical equilibrium is assessed as follows: the relative frequency of each master sequence is tested for the condition of $\geq 10^{-3}$ (cut-off relative frequency for survival) in every 2000th replication step in the interval between 10,000 replication steps before termination and actual termination ($10^7$ replication steps, unless stated otherwise). According to our numerical experiences, convergence to an apparent steady state is always reached until the beginning of this time interval, even though stochastic extinction events can occur at any time. If the above condition for the cut-off relative frequency for survival holds in more than one such sample, the master type in question is regarded as survived, otherwise it is considered to be extinct in dynamical equilibrium. All simulations were performed in C. Statistical tests included in this manuscript were performed in the R environment (*R Development Core Team, 2023*), using two-tailed hypothesis tests with alpha risk of 0.05.

## Results

### Constant population model

#### Coexistence of the replicators

First, we investigated the diversity-maintaining ability of the parabolic dynamics at different population sizes and replicability distances. *Figure 2A* shows the results of individual runs with $T = 10$ master types in case of small ($N = 10^3$), medium ($N = 10^4$), and large ($N = 10^5$) population size and

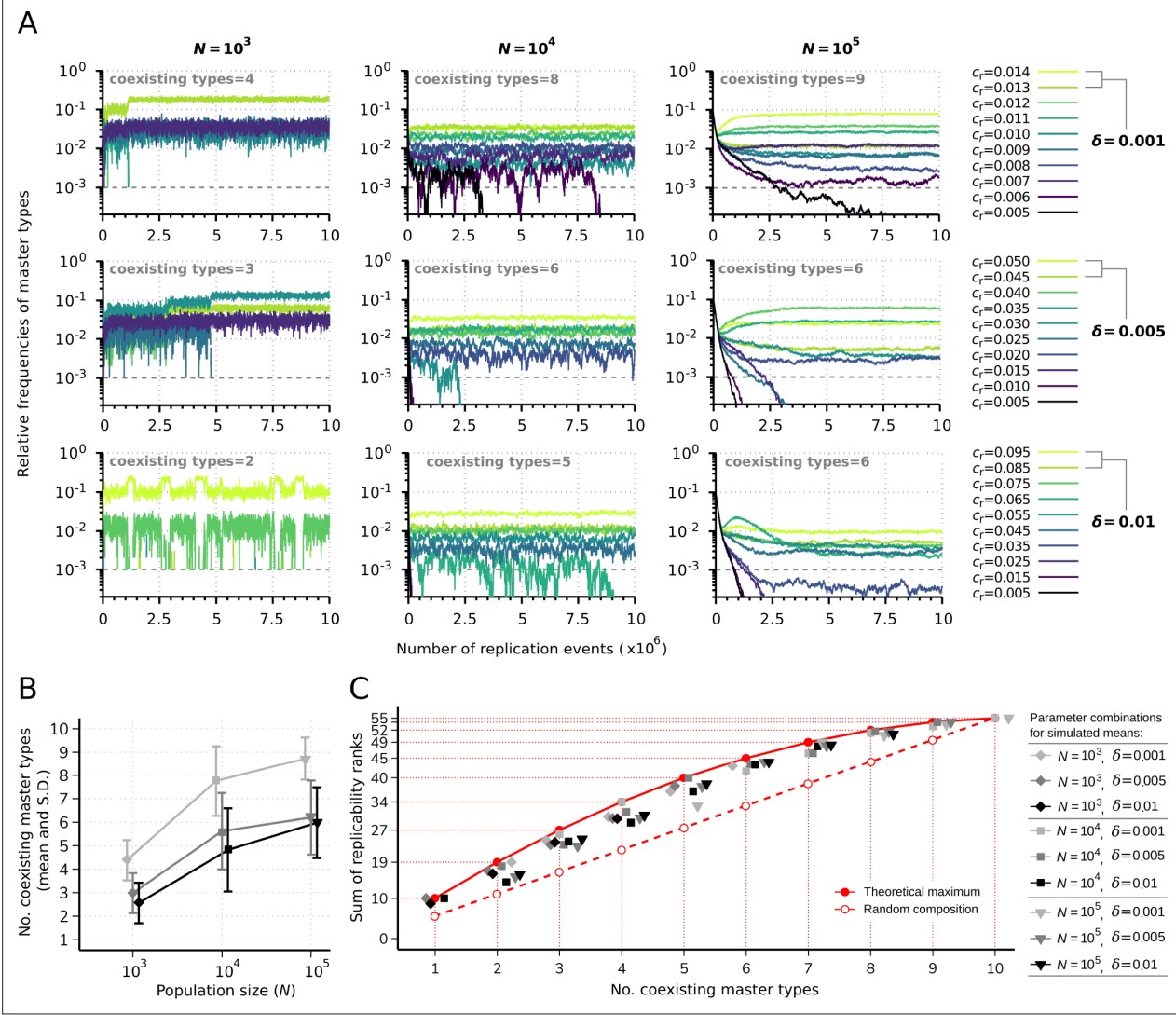

**Figure 2.** Replicator coexistence in the constant population model. (**A**) Time series of the relative frequencies of master types. Panels corresponding to different rows demonstrate distinct replicability distances ($\delta$), panels corresponding to different columns demonstrate distinct population sizes ($N$). Line colors indicate different replicabilities of the master types (as explained in the legends). The cut-off relative frequency for survival is indicated by horizontal dashed lines. (**B**) Mean and standard deviation of the number of sustained master types as the function of $N$ and $\delta$ calculated from 50 independent runs (the corresponding legend is shown next to panel **C**). (**C**) Sum of replicability ranks of the survived master types at different number of coexisting types, corresponding to the simulation results shown in (**B**). The red curve denotes the maximum of the sum of replicability ranks ($\rho_{max}$), the dashed red line shows the expected values of the sum of replicability ranks, when the surviving master types are random ($\rho_{rand}$) with respect to their replicabilities. The default parameter set (see *Table 1*) was used, unless otherwise indicated.

small ($\delta = 0.001$), medium ($\delta = 0.005$), and large ($\delta = 0.01$) replicability distances. In all cases, the smallest replicability of the master types is $\phi_1 = \phi_{min} = 0.005$. Note that, the ratio of the smallest and largest master replicabilities $\left(\frac{\phi_{10}}{\phi_1}\right)$ in the cases of three different $\delta$ s are 2.8, 10, and 19, respectively. These enormously large fitness ratios and fitness differences result in a highly competitive regime in Darwinian sense and also provide the opportunity to investigate the diversity-maintaining capacity of the parabolic regime. The other key parameter of the system is the population size. It is known that in the infinite population size limit, the sustainable diversity has no upper bound (*Szathmáry and Gladkih, 1989*), but we are interested in how the sustained diversity decreases with decreasing population size. As a general observation, we can state that with decreasing population size, the effect of demographic stochasticity increases and can drive some of the master types extinct. *Figure 2A* shows that increasing the population size reduces the fluctuations in the frequencies and a higher

population size together with smaller replicability distance increases the number of sustained master types. We measured the average and standard deviation of the number of maintained master types as the function of population size and replicability distance (*Figure 2B*). In case of small population size, the parabolic regime can maintain 43.8%, 29.8%, and 25.6% of the master types at an average, assuming small, medium, and large replicability distance, respectively. The corresponding percentages of sustained master types in case of medium population size are: 77.6%, 56.2%, and 48.2%; and in case of large population size: 87.2%, 62%, and 59.8%. The average values are also in agreement with the expectations: both large population size and small replicability distance increase the sustainable diversity.

A set of survived master replicator types and thus the selectivity of a parabolically replicating system can be characterized by the sum of replicability rank indices of the coexisting master population. According to *Equation 8*, the master type with the highest replicability has rank 10, the second highest has rank 9, while the master type with the lowest replicability has rank 1. If $T_{surv}$ different types of master replicators survive (coexist), the maximum of the sum of their ranks is

$$\rho_{max} = \sum_{i=1}^{T_{surv}} 11 - i = \frac{T_{surv} \cdot (21 - T_{surv})}{2} \, , \, (1 \leq T_{surv} \leq T).$$ The expected value of the sum of ranks, when the surviving masters are distributed randomly with respect to their replicabilities, is $\rho_{rand} = 5.5 \cdot T_{surv}$.

According to our simulation results, the sum of replicability ranks of the coexisting types is typically

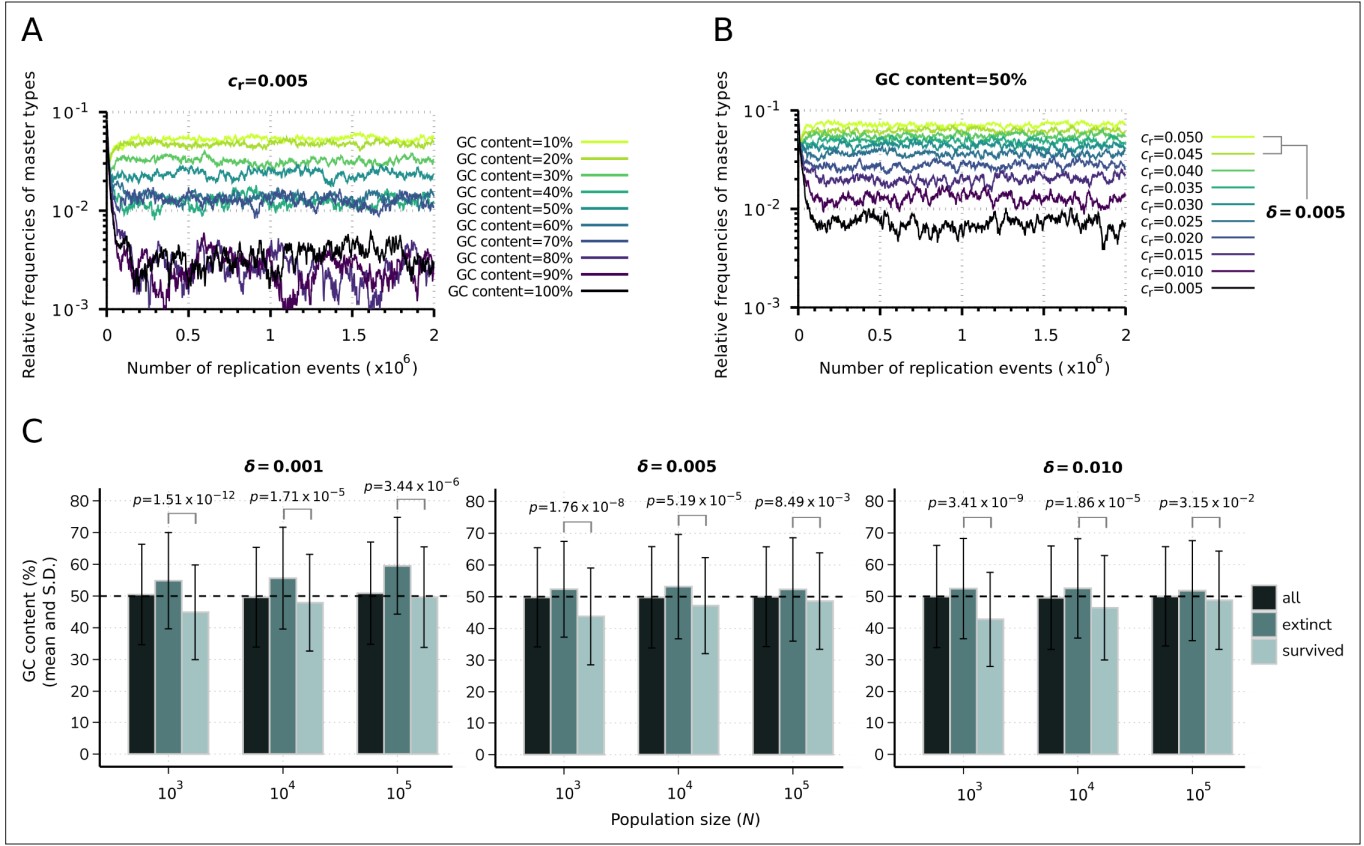

**Figure 3.** Effect of sequence composition on replicator abundances in the constant population model. (**A**) Demonstration of the direct effect of sequence composition on replicator abundances. The master types (indicated by differently colored lines) are defined along a GC content gradient with identical replicabilities ($c_r = 0.005$). (**B**) Consequences of removal of the effect of sequence composition on replicator abundances. The master types (indicated by differently colored lines) are defined along a replicability gradient (replicability distance: $\delta = 0.005$) with identical GC contents (50%). Due to the relatively fast convergence, simulations were terminated after $2 \cdot 10^6$ replication events. Parameters that were used in both simulations: $N = 10^4$, $p_{mut} = 0.01$ (see *Table 1*). (**C**) Means and standard deviation of the relative GC content in the survived and extinct master types, corresponding to the simulation results shown in *Figure 2B and C*. Columns with the darkest shade represent the average GC content of the randomly generated $n = 500$ master sequences (10 sequences for all 50 replicate runs) constituting the initial populations. Horizontal dashed lines indicate the expected (random) 50% initial GC content. The exact *p*-values resulted from the two-sample Wilcoxon tests on the GC content of the survived-extinct subsets are indicated above the column pairs.

lower than the theoretical maximum ($\rho_{max}$), but higher than the random composition ($\rho_{rand}$), indicating that the fitness of a replicator does not only depend on its replicability and therefore the dynamics does not *only* selects for relatively high replicabilities (*Figure 2C*).

### Effect of sequence composition

We found that the key factor leading to the deviation from the theoretical maximums of the sum of replicability ranks of the surviving master subsets is the GC content (the sum of G and C nucleobases) of the sequences. The direct effect of GC content on the replicator dynamics and thus the selectivity of a parabolically replicating system with regard to sequence composition were investigated by observing the time evolution of $T = 10$ master sequences generated along a GC content gradient assuming equal replicabilities. For this investigation, we allocated the same number of G+C nucleobases (with 50–50% probability of G and C) to the master types as their indexes. Then A or U nucleobases (also with 50–50% probability) were allocated to the remaining loci and random shuffling of the nucleobases among the loci was carried out. This analysis showed that equilibrium master sequence frequencies follow an inverse relation with the GC content gradient, with increasing stochastic fluctuations in the frequencies at relatively large GC content values (*Figure 3A*).

To address the question of how the parabolic dynamics operates when the above-described effect of sequence composition is excluded, we examined master sequence abundances under the assumption of balanced GC content, along a replicability gradient and with cross-hybridization between different master types not being allowed. In this scenario, master sequences were generated in the following way: equal GC content of the sequences was ensured by allocating G or C (with 50–50% probability) nucleobases to the first half of the loci. Then, A or U nucleobases (also with 50–50% probability) were allocated to the remainder second half of the loci and random shuffling of the nucleobases among the loci was carried out. Another restricting condition considered during this investigation is that master-master and complementary-complementary sequence associations are not allowed, nor master-complementary hybridization of different types, thereby preventing the formation of cross-hybrid duplexes with different binding energies. This investigation showed that if the effects of sequence composition and cross-hybridization are excluded, (i) equilibrium master sequence frequencies are directly proportional to their replicabilities and exactly follow the replicability gradient, (ii) survival of each master sequence is ensured because of the fact that in this way

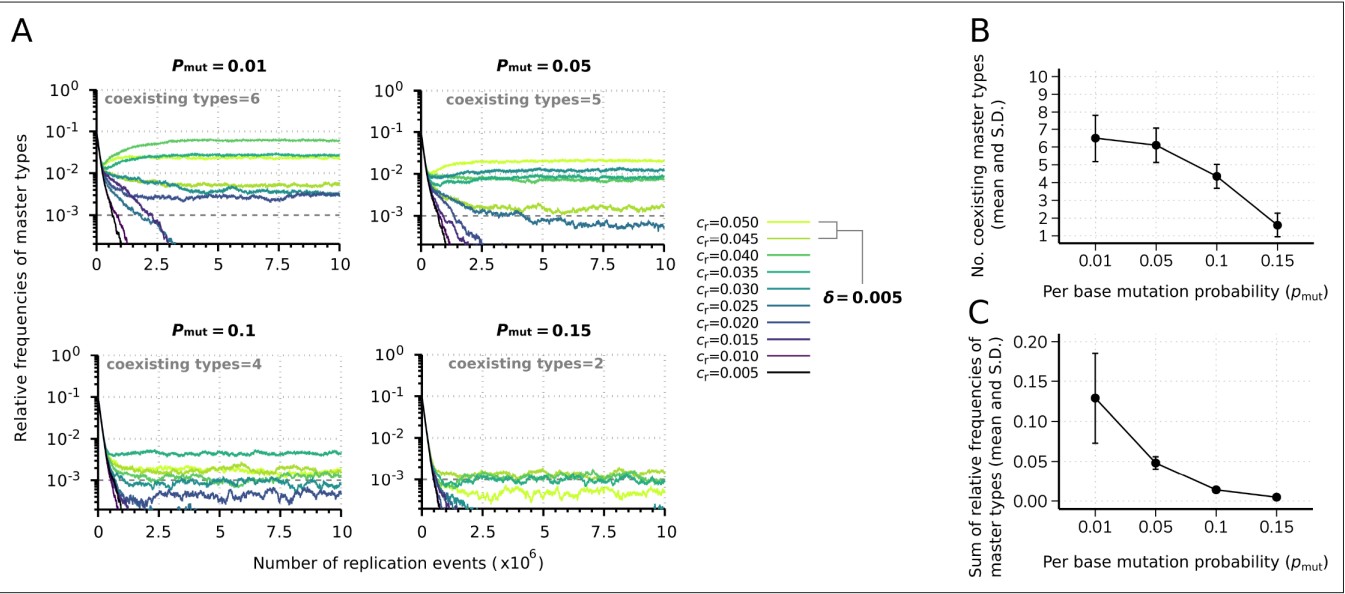

**Figure 4.** Effect of the mutation rate on the replicator coexistence in the constant population model. (**A**) Time series of the relative frequencies of master types at different mutation rates ($p_{mut}$). Line colors denote master type replicabilities (values are indicated in the legend). The cut-off relative frequency for survival is indicated by horizontal dashed lines. (**B**) Mean and standard deviation of the coexisting master types as the function of $p_{mut}$ calculated from 20 independent runs. (**C**) Mean and standard deviation of the sum of relative frequencies of master types in the total population as the function of $p_{mut}$, corresponding to the simulation results shown in (**B**). Except for $p_{mut}$, the default parameter set (see *Table 1*) was used.

each master inhibits (regulates) only itself and no other masters, i.e., the dynamics, is 'purely' parabolic (**Meszéna and Szathmáry, 2002**), (iii) despite a relatively low total population size ($N = 10^4$), stochastic fluctuations have negligible effects on the dynamics (**Figure 3B**). A direct investigation of whether the sequence composition of the master types is associated with their survival outcome was conducted using the data from the constant population model simulation results (**Figure 2**). In these data, the average GC content was measured to be lower in the surviving master subpopulations than in the extinct subpopulations (**Figure 3C**).

To determine whether this difference was statistically significant, nonparametric, two-sample Wilcoxon rank-sum tests (**Hollander and Wolfe, 1999**) were performed on the GC content of the extinct-surviving master subsets. The GC content was significantly different between these two groups in all nine investigated parameter combinations of population size ($N$) and replicability distance ($\delta$) at $p < 0.05$ level, indicating a selective advantage for a lower GC content in the constant population model context. The exact $p$-values obtained from this analysis are shown in **Figure 3C**.

## Effect of mutations

We investigate the effect of the mutation rate on the sustainable diversity and the proportion of master types in the total replicator population. **Figure 4A** shows four realizations with different per bit mutation rates: $p_{\mathrm{mut}} = 0.01, 0.05, 0.1, 0.15$. In accordance with the expectations, increasing mutation rates reduce the equilibrium abundances of masters by widening the mutant cloud. **Figure 4B** shows this effect as average of 20 independent runs for each mutation rate. The corresponding average of percentages of sustained master types are: 65%, 61%, 43.5%, and 16%, respectively. Summed relative steady-state master frequencies are a decreasing function of the mutation rate, see **Figure 4C**.

## Lack of error threshold in case of parabolically replicating infinite populations

As a reference case, we analyzed the behavior of a simplified dynamics of parabolic replication (**Equation 15A** and **Equation 15B** in Appendix 1). For analytic tractability, we assume infinite population size and a single peak replication landscape in which a single genotype has high replication rate, all others have the same baseline value, the ratio of these two values (denoted by $A$) is the reproductive superiority. Furthermore, we consider the essential requisite that ensures parabolic dynamics, namely that the growth rate of the population is proportional to the square root of the actual size of the population.

Our analytical results show that in this simple model – except for $Q = 0$, where $Q$ denotes the probability of the error-free replication of a master sequence – there is no error threshold: the master type will persist at any level of mutation rate (**Figure 5**, right panel). By contrast, in non-parabolic dynamics (i.e. in the Eigen model, referred to as exponential dynamics, see, **Equation 14A** and **Equation 14B** in Appendix 1), the master type disappears from the system above a critical mutation rate $Q^* = \frac{1}{A}$ (**Figure 5**, left panel).

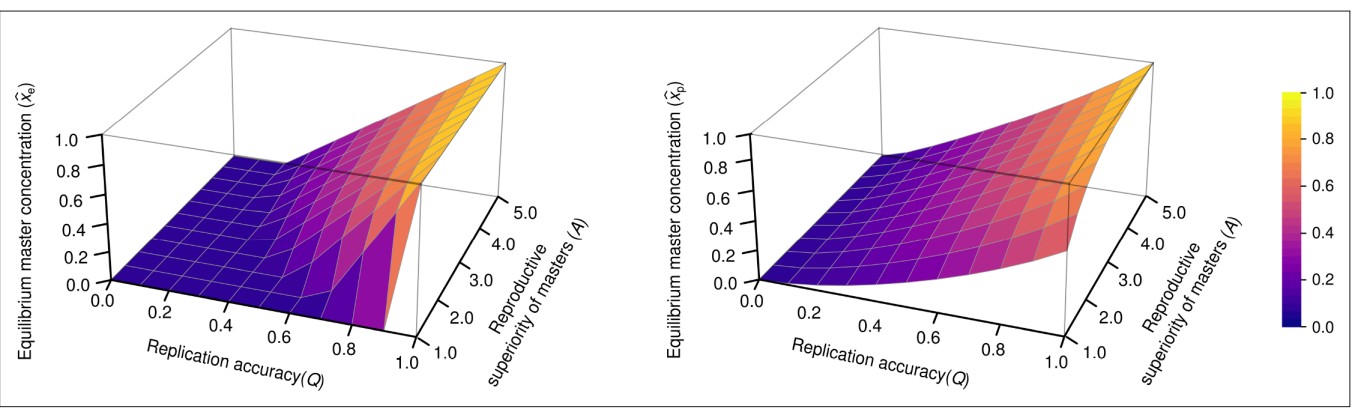

**Figure 5.** The concentration of master type ($\hat{x}_{\mathrm{e}}$ and $\hat{x}_{\mathrm{p}}$) as the function of $Q$ and $A$ in the exponential model (left panel) and in the parabolic replication model (right panel). $a = 1$ (where $a$ stands for the reproductive value of the mutants) was used.

Introducing back mutations into the model (so far neglected as a rare process) fundamentally changes the dynamics: the error threshold also disappears from the exponential regime, see *Appendix 1—figure 2*. In the parabolic case, under a realistic parameter range – e.g., in the interval, where the probability of back mutation $p_B = [0; 0.1]$ – the smaller the probability of back mutations the larger the equilibrium concentration of the parabolic masters in comparison with that of the exponential masters (*Appendix 1—figure 3D*). In terms of the difference in the equilibrium concentrations between the parabolic and the exponential masters, the smaller the replication fidelity, the higher the relative parabolic master concentration for both mutation scenarios (*Appendix 1—figure 1* and *Appendix 1—figure 3A–C*).

## Chemostat system

### Diversity-maintaining ability

The chemostat model, by taking into account the components of replicator decay, inflow of nucleobases into the system, and outflow of the molecules, shows a richer dynamical repertoire compared to the constant population model. The total population size and the diversity-maintaining ability of the system are jointly affected by both the inflow rate of the nucleobases ($c_{in}$) and the decay rate of replicators in simplex ($c_{\delta_{ss}}$) and duplex ($c_{\delta_{ds}}$) form, *Equation 11*. *Figure 6A* shows individual realizations of this dynamics with the parameter values that lead to comparable equilibrium population sizes to those investigated in the constant population model (*Figure 2*). Note that in the chemostat model, the total population size ($N$) changes in time and therefore population sizes shown in *Figure 6A* are approximate measures. Consistent with the constant population model, the dynamical outcome approaches complete coexistence if large population sizes are combined with relatively similar replicabilities (*Figure 6A*, upper right panel) under the assumption of small duplex decay rates relative to that of the simplexes ($f = 0.01$). In line with the expectations, decreasing $c_{in}$ reduces the average total population size $\overline{N}$, and this simultaneously decreases the number of sustainable master types (*Figure 6AB*). Our results show that decreasing $c_{in}$ by one order of magnitude, as expected, leads to approximately one order of magnitude drop in the average total population sizes (*Figure 6B*). Relatively large replicability distance parameter ($\delta$) values, however, while narrowing down the range of the coexistence, increase the average total population sizes at the same time (*Figure 6B*) due to the fact that they imply larger mean master replicabilities.

For further analysis, we introduce the simplex-duplex asymmetry measure: $\chi = \frac{N_{simplex} - N_{duplex}}{N}$, whose negative values indicate duplex, positive values indicate simplex dominance, –1 (+1) value means that the whole population is in duplex (simplex) state. Along an increasing gradient of the duplex decay factor ($f$), the average total population sizes are again lowered (*Figure 6B*). The mechanistic explanation for this population-shrinking effect at relatively large values of $f$ is well demonstrated by the population compositions, showing that the overwhelming majority of the replicators tend to be in double-stranded state independent from $\delta$ (*Figure 6C*). Note, however, the considerable shift toward the single-stranded state along a decreasing resource influx ($c_{in}$) and average total population size gradient (*Figure 6C*).

### Shift in selectivity

In case of a strongly resource-limited regime, the diversity-maintaining potential of the dynamics can ultimately drop to a level at which the system can maintain only one master type (*Figure 6B*). Our results show that such a single-species equilibrium generally occurs under the assumption of the smallest resource inflow rate ($c_{in} = 0.01$), combined with large replicability distance ($\delta = 0.01$) and duplex decay factor ($f = 1$). In the chemostat model binding energy of a duplex (and thus its sequence composition) is also taken into account in the decay probability distribution (*Equation 11*). Therefore, a higher GC content results in a lower decay rate, compensating to some extent the otherwise disadvantageous effect of a relatively high GC content (*Figure 3A* and *Figure 3C*) and the consequential low dissociation probability (only dissociated, single-stranded sequences are available as templates for complementary strand synthesis). This sequence composition effect-mediated trade-off between lowered decay rate in duplex state and templating affinity in simplex state, together with type-specific replicabilities, determines the identity of the surviving master types and thus the selectivity of the system in single-species equilibria.

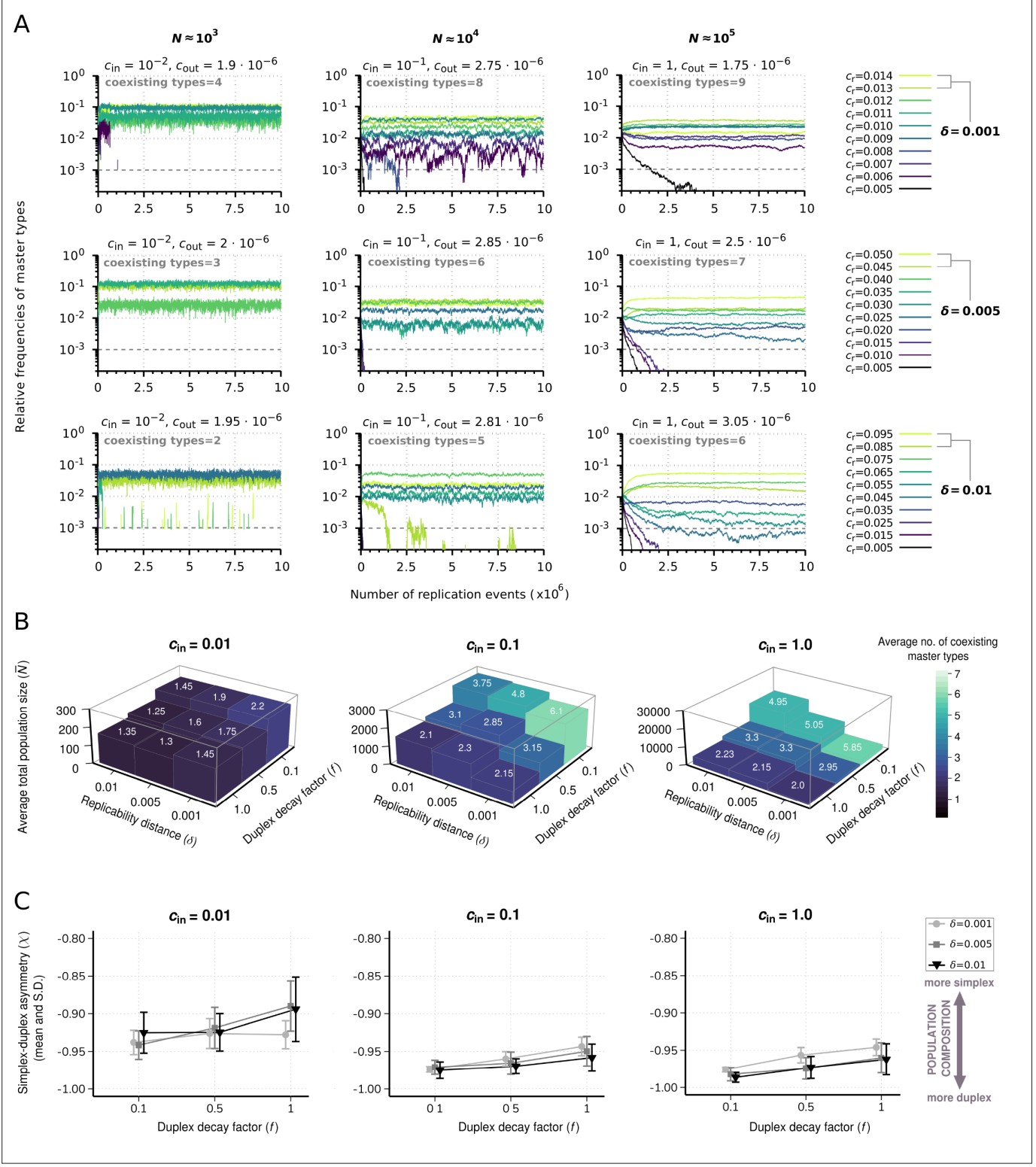

**Figure 6.** Effect of resource competition on replicator coexistence in the chemostat system model. (**A**) Time series of the relative frequencies of master types. Panels corresponding to different columns demonstrate different population sizes ($N$). Panels corresponding to different rows demonstrate different replicability distances ($\delta$). Line colors indicate master types with distinct replicabilities (as explained in the legends). The cut-off relative frequency for survival is indicated by horizontal dashed lines. (**B**) Averages for the total population size ($\bar{N}$) and the number of coexisting master types (color and numbers on columns) as functions of $\delta$ and duplex decay factor ($f$) at different resource inflow rates ($c_{in}$). Averages were obtained from 20

*Figure 6 continued on next page*

*Figure 6 continued*

individual runs for every parameter combination. (**C**) Mean and standard deviation of simplex-duplex asymmetry $\left(\chi = \frac{N_{simplex} - N_{duplex}}{N}\right)$ as the function of $\delta$ and $f$ at different $c_{in}$ values, corresponding to the simulation results shown in (**B**). The default parameter set (see *Table 1*) was used, unless otherwise indicated.

## Discussion

In this modeling study we focused on the coexistence of polymer variants involved in a template-directed, non-enzymatic replication mechanism. Prior to the emergence of a helicase ribozyme that could have facilitated strand separation in RNA duplexes (or a similar, relatively effective strand separation mechanism, see, for example, *Szostak et al., 2001*; *Müller, 2006*), such an enzyme-free replication mode was likely to result in parabolic growth profiles (*von Kiedrowski et al., 1991*). Therefore, parabolic population growth is of conceptual relevance to the origin and the subsequent evolution of heritable chemical information. Within this framework, we demonstrated that – depending on the total replicator population size, GC content, the extent of replication rate differences and copying error, as well as resource influx rates – parabolic dynamics can sustain a large variety of sequence modules.

A small total population size ($10^3$) combined with large replication rate differences ($\delta$=0.01), on average, can only maintain 26% of all species (*Figure 2B*). These results corroborate Davis's (2000) analytical findings that steady-state replicator variant frequencies in a finite, sublinear system are truncated in such a way that variants with insufficient fitness are eliminated. In contrast, large population sizes and smaller between-species fitness differences are expected to approach the deterministic, unlimited coexistence limit (*Szathmáry and Gladkih, 1989*; *Varga and Szathmáry, 1997*). This is confirmed here by considering the combination of a relatively large total population size ($10^5$) and small equidistant difference in the replication rates ($\delta$=0.001), which leads to the coexistence of almost 90% of the variants, on average (*Figure 2B*). The replicability (fitness) distance dependency of the number (or the proportion) of the coexisting variants in dynamical equilibrium (*Figure 2B*) is viewed as the manifestation of fitness-driven (Darwinian) selection during competitive replication at sublinear propagation rates (*Davis, 1996a*; *Davis, 1996b*). Analytical and numerical results show that the threshold fitness for coexistence depends on the fitness sum in such a system (*Davis, 2000*). Consistent with this, elevation in the sum of species fitness – that can be expressed in the present model in terms of the increasing sum of replication rates of master species with increasing replicability distance ($\delta = 0.001 : \Sigma_{i=1}^{T} c_{r(i)} = 0.095; \delta = 0.005 : \Sigma_{i=1}^{T} c_{r(i)} = 0.275; \delta = 0.01 : \Sigma_{i=1}^{T} c_{r(i)} = 0.5$) – raises the fitness threshold for survival and thus the number of subthreshold species that are eliminated (*Figure 2A, B*) due to the fact that master species with higher fitness are present in higher copy numbers (*Figure 2A*).

With regard to demographic stochasticity, we considered order of magnitude-scaled distances between the investigated population sizes, while a constant cut-off relative frequency for survival ($10^{-3}$) was applied, implying that the minimum copy number to avoid exclusion also increased by an order of magnitude with increasing population size (*Figure 2A*). The rationale behind this assumption is that in this way the average number of the maintained variants is expected to change linearly as a function of population size, if fluctuations associated in population size shifts affect the diversity-maintaining potential linearly. In contrast to linear dependence, however, we found that the slope of this function is steeper between the smallest and the medium population sizes than between the medium and the largest one and that this non-linearity is becoming more pronounced with decreasing fitness differences (i.e. with smaller $\delta$, *Figure 2B*). Accelerating species extinction at the smallest population size, resulting from increased variability in relative species frequencies over time (i.e. increased amplitude of stochastic fluctuations), is consistent with prior predictions and analytical findings that suggested the existence of a critical total replicator population size above which parabolic coexistence is guaranteed (*Epstein, 1979*; *Scheuring and Szathmáry, 2001*). These remarks emphasized that effective diversity maintenance by parabolic replication should have required sufficiently large population sizes during early molecular evolution, which, according to our results, must have been >$10^3$.

Moreover, we found evidence that under sublinear dynamics, GC content of the replicator sequences has also a significant effect on the steady-state master frequencies (*Figure 3A*) and, thus, on the identity of the surviving master types (*Figure 2C* and *Figure 3C*). Consequently, this feature also plays a decisive role in determining the number of the surviving variants in equilibrium in a manner

that a balanced GC content favors a larger number of coexisting variants (*Figure 3B*), whereas, for example, an extremely low GC content of a few master sequences is expected to lead the survival of this narrow subpopulation of the sequences with low GC content. This finding can be interpreted in the light of the lifetime reproductive ratio of the replicators. Reproductive ratio in this case can be defined as the expected number of copies ('offspring') produced by a single strand or, in case of complementary template replication, the number of complementary strands synthesized from a given sequence during its lifespan (*Meszéna and Szathmáry, 2002*). According to this interpretation, a relatively high dissociation probability and the consequential higher propensity of being in single-stranded form provides an advantage to the sequences with relatively low GC content in terms of their replication affinity, i.e., the expected number of offspring in case of such variants will be relatively high. This model characteristic is also in accordance with the results of a computational study on RNA folding from randomly generated sequences, showing that abundant and topologically simple fold structures tend to arise from sequences depleted in G, thereby suggesting that the first catalytic RNA sequences could have been characterized by a somewhat reduced GC content (*Briones et al., 2009*).

We demonstrated that during competitive replication of parabolically growing replicator variants, monomer resource inflow rate ($c_{in}$) can act as a control parameter of selectivity. As such, by changing the values of this parameter, the dynamics can switch from a non-Darwinian steady state that ensures coexistence toward a Darwinian state, where one competitively superior species prevails, other variants are competitively excluded. In an analogous replicator competition model, Szilágyi and co-workers applied Gause's principle of competitive exclusion (*Gauze, 1934*) to replicator populations and showed both analytically and numerically that if resources (nucleotides) affect the growth linearly, then the number of coexisting replicator species cannot be more than that of the nucleotide building block types (note that plus and minus copies of the same replicator count as one species: *Szilágyi et al., 2013*). By contrast, the present study provides evidence that within the context of sublinear propagation, at high monomer resource concentrations that ensure large population sizes (*Figure 6A*), the average number of the maintained replicator types can be higher than the number of resource types (4) (*Figure 6B*). However, with small monomer inflow rate, sublinear population growth also becomes resource limited concomitant with a considerable drop in the average total population size (*Figure 6B*), leading to the survival of the fittest variant.

We also investigated the consequences of errors in the replication mechanism in the context of the theoretical error threshold of parabolic dynamics. Our analytical results show that a master sequence vanishes from the system, only if the replication accuracy is zero (*Figure 5*, right panel). Thus, in this deterministic scenario, the error threshold problem is seemingly circumvented, because persistent maintenance of essential genetic information by non-enzymatic, template-directed synthesis of short RNA-like information carrying sequence modules is not an irresolvable task for a putative rudimentary replication mechanism, even if it is highly inaccurate. However, when finite population sizes and stochastic effects are taken into account, at the largest investigated per-base mutation rate ($p_{mut} = 0.15$), the summed relative steady-state master frequencies approach zero (*Figure 4C*) with accelerating species extinction (*Figure 4B*), indicating that this value is close to the system's empirical error threshold. We note that this value is above the one that can be anticipated if one applies the inverse genome length rule of thumb for the error threshold (*Eigen, 1971*; *Joyce, 2002*) to our system, which gives $1/L = 0.1$ error rate of replication per nucleotide (but note that applying this approach to our system is a serious oversimplification). Moreover, the applied constant cut-off relative master frequency for survival ($10^{-3}$) renders our empirical error threshold approximation rather conservative. Former studies suggest that a 0.1 per-base error rate, at which we measured a decent diversity-maintaining potential (more than 40% of all variants), is a reasonable assumption for non-enzymatic RNA copying. Although the average error rate in this process was estimated to be ~0.17 (*Leu et al., 2011*), using optimized nucleotide ratios could potentially lower this measure below ~0.1 with a possibility for a further reduction to ~0.05 on GC-rich templates (*Szostak, 2012*). However, all of the average error rates reported by these studies pertain to non-enzymatic polymerization reactions involving primers. Triplet substrates, albeit in an enzymatic context (i.e. with in vitro evolved ribozymes), have proven to be able to successfully bypass the requirement of primers (*Attwater et al., 2018*). A similarly primer-free and therefore – under prebiotic conditions – more realistic replication mode has been proposed for non-enzymatic RNA replication in which complementary strand synthesis takes place by oligonucleotide ligation together with gap filling reactions,

making use of monomers and shorter oligomers (*Szostak, 2012*). Although the fidelity of this RNA replication mechanism is largely unknown, a pilot experimental study (*James and Ellington, 1997*), which was performed with constant-sequence DNA templates, suggests a surprisingly high accuracy of complementary strand synthesis during this process.

## Ideas and speculation

As we have demonstrated, template-directed, non-enzymatic replication of oligomer modules leading to parabolic growth profiles is an effective genetic information maintaining mechanism which may thus have constituted an indispensable bridge from the very first abiotic synthetic pathways for RNA to a ribozyme-dominated stage of the RNA world. The demonstrated diversity-maintaining mechanism of finite parabolic populations can be used as a plug-in model to investigate the coevolution of naked and encapsulated molecular replicators (e.g. *Babajanyan et al., 2023*). It is, however, noteworthy that besides the 'information first' view of the RNA world hypothesis, there are other concepts as to how life arose; e.g., the 'metabolism first' or autotrophic origins view (*Wächtershäuser, 1988*; *Joyce, 2002*; *Martin et al., 2008*). Moreover, this view does not exclude the possibility of the existence of a pre-RNA world, in which genetic information resided in polymer structures similar to RNA, such as peptide or tetrose nucleic acids (*Schöning et al., 2000*; *Bada, 2004*; *Orgel, 2004*). Considering that we applied a rather broad approach to replicator representation we believe that the results of the present model are general enough for being naturally applicable to pre-RNA systems as well. Although short oligonucleotides like those we have investigated in the present study are not characteristically prone to intramolecular 2D folding and therefore typically bear with limited catalytic activities, miniribozymes with surprisingly short sequence lengths can still catalyze a wide variety of chemical reactions (*Vlassov et al., 2005*). Therefore, catalytic aid and/or metabolic cooperation among independently replicating sequences (i.e. *cooperative coexistence*, see, for example, *Epstein, 1979*; *Mizuuchi and Ichihashi, 2018*) can considerably broaden the parameter space in which we reported *competitive coexistence* of different replicator variants.

In this study we considered kinetic constants that correspond to the $p \approx 0.5$ growth order. Further investigations into the $0.5 < p < 1$ interval (*Issac and Chmielewski, 2002*; *Li and Chmielewski, 2003*; *Lincoln and Joyce, 2009*) are expected to show that an increase in the kinetic growth order decreases the scope of competitive coexistence and drives the system more toward the Darwinian regime. Such studies may help gain a deeper insight into the question whether the emergence of an effective strand separation mechanism, or other conditions implying kinetic constants that shift the dynamics toward exponential amplification, allowed for a still sub-exponential diversity-maintaining mechanism during later phases of chemical evolution and thus a gradual evolutionary transition from non-enzymatic replication to ribozyme-aided amplification.

The two key components of evolutionary algorithms are exploration and exploitation (*Bäck et al., 2000*). If diversity decreases too fast under selection, this can lead to premature convergence, resulting in a population getting stuck at a local adaptive peak. In order to avoid premature convergence, various diversity-maintaining mechanisms have been introduced. We call attention to the fact that parabolic growth is an organic form of diversity maintenance. Referring to the chemostat model, we propose that a periodically changing environment with alternating resource abundance and shortage can drive, and in the distant past could have driven, an efficient evolutionary exploration-exploitation algorithm.

## Acknowledgements

This work was supported through National Research, Development and Innovation Office Élvonal KKP129848 and the Templeton World Charity Foundation ('Learning in evolution, evolution in learning' award - TWCF0268). AS received support from the Hungarian Academy of Sciences through Bolyai János Research Fellowship program. MP received support from the ELTE Eötvös Loránd University through Hungarian state PhD scholarship. We thank Géza Meszéna for helpful discussions, as well as János Podani for comments on the manuscript. We would like to thank Balázs Könnyű for his advice on the statistical analysis and Dániel Vörös for his help with the coding.

# Additional information

## Funding

| Funder | Grant reference number | Author |
| --- | --- | --- |
| National Research, Development and Innovation Office | Élvonal KKP129848 | Mátyás Paczkó Eörs Szathmáry András Szilágyi |
| Templeton World Charity Foundation | TWCF0268 | Mátyás Paczkó Eörs Szathmáry András Szilágyi |
| Hungarian Academy of Sciences | Bolyai János Research Fellowship | András Szilágyi |
| Eötvös Loránd University | Hungarian state PhD scholarship | Mátyás Paczkó |

The funders had no role in study design, data collection and interpretation, or the decision to submit the work for publication.

## Author contributions
Mátyás Paczkó, Conceptualization, Software, Visualization, Writing – original draft, Writing – review and editing; Eörs Szathmáry, Conceptualization, Formal analysis, Funding acquisition, Methodology, Writing – original draft, Writing – review and editing; András Szilágyi, Conceptualization, Formal analysis, Supervision, Funding acquisition, Visualization, Methodology, Writing – original draft, Writing – review and editing

## Author ORCIDs
Mátyás Paczkó ⓘ https://orcid.org/0000-0002-0162-7927
Eörs Szathmáry ⓘ http://orcid.org/0000-0001-5227-2997
András Szilágyi ⓘ http://orcid.org/0000-0002-6894-4652

Reviewer #1 (Public review): https://doi.org/10.7554/eLife.93208.3.sa1
Reviewer #2 (Public review): https://doi.org/10.7554/eLife.93208.3.sa2
Author response https://doi.org/10.7554/eLife.93208.3.sa3

# Additional files

## Supplementary files
• MDAR checklist

## Data availability
The current manuscript is a computational study, so no data have been generated for this manuscript. Modelling code is publicly available on GitHub under the GNU General Public License v3.0 license: https://github.com/paczkomatyas/Parabolic_Replicator_2023 (copy archived at *Paczkó, 2024*).

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

# Appendix 1

## Simple Maynard-Smith-type error threshold for parabolic replicators

### No back mutation

The dynamics of the master type (denoted by $x$) and the mutant types ($y$) for the exponential regime in the original Maynard-Smith model (**Maynard Smith and Szathmáry, 1995**; pp. 44–49) are as follows:

$$\dot{x} = AQx - x\left(Ax + ay\right) \tag{14A}$$

$$\dot{y} = ay + A\left(1 - Q\right)x - y\left(Ax + ay\right), \tag{14B}$$

while its parabolic counterpart has the following form:

$$\dot{x} = AQ\sqrt{x} - x\left(A\sqrt{x} + a\sqrt{y}\right) \tag{15A}$$

$$\dot{y} = a\sqrt{y} + A\left(1 - Q\right)\sqrt{x} - y\left(A\sqrt{x} + a\sqrt{y}\right), \tag{15B}$$

where $A$ and $a$ the master and mutant reproductive values, respectively, if $0 \leq Q \leq 1$ is the probability of the error-free replication of a master sequence, the last terms in the equations guarantees the constant unit concentration ($x + y = 1$).

The non-trivial fixed points of **Equation 14A** and **Equation 14B** are:

$$\left(\hat{x}_0, \hat{y}_0\right) = \left(0, 1\right) \tag{16A}$$

$$\left(\hat{x}_e, \hat{y}_e\right) = \left(\frac{AQ - a}{A - a}, \frac{A\left(1 - Q\right)}{A - a}\right). \tag{16B}$$

If $Q > \frac{a}{A}$ the $\left(\hat{x}_e, \hat{y}_e\right)$ fixed point is asymptotically stable, the master type can coexist with its mutants, $Q^* = \frac{a}{A}$ is the error threshold where this fixed point loses stability and when $Q < \frac{a}{A}$ the $\left(\hat{x}_0, \hat{y}_0\right)$ becomes the stable fixed point, the master type dies out.

In contrast to the exponential system, parabolic dynamics does not exhibit a sharp change in the concentration of the master as the function of the replication fidelity $Q$. The interior fixed point of the system (which is asymptotically stable in the $0 < Q < 1$ region):

$$\hat{x}_p = \frac{a^2 + 2A^2Q - \sqrt{a^4 + 4a^2A^2Q\left(1 - Q\right)}}{2\left(a^2 + A^2\right)} \tag{17A}$$

$$\hat{y}_p = \frac{a^2 + 2A^2 - 2A^2Q + \sqrt{a^4 + 4a^2A^2Q\left(1 - Q\right)}}{2\left(a^2 + A^2\right)}. \tag{17B}$$

It is easy to prove that $\hat{x}_p$ is always positive in $0 < Q < 1$, since the expression under the root sign can be estimated as follows:

$$\sqrt{a^4 + 4a^2A^2Q\left(1 - Q\right)} = \sqrt{\left[a^2 + 2A^2Q\left(1 - Q\right)\right]^2 - 4A^4Q^2\left(1 - Q\right)^2} < a^2 + 2A^2Q.$$

Note that the master type vanishes if and only if $Q = 0$ (**Figure 5**, right panel), and mutants cannot vanish because of parabolic competition. The master type can be lost only through stochastics at sufficiently low equilibrium $\hat{x}_p$ values, but not otherwise.

When $Q^{**} = \frac{a + A}{2A}$, both regimes have the same master concentrations ($\hat{x}_p = \hat{x}_e$), below this value the parabolic, above this value the exponential system has more masters. As $Q^{**} > Q^*$ (whenever $A > a$), the length of the replication accuracy interval where the parabolic regime maintains a higher equilibrium master concentration ($Q^* < Q < Q^{**}$):

$$Q^{**} - Q^* = \frac{A - a}{2A}. \tag{18}$$

This means that the 'parabolic higher' $Q$ range increases with the reproductive superiority of masters ($A$). **Appendix 1—figure 1** shows the different outcomes as the function of the $A$ and $Q$, the continuous red and dashed black curves correspond to $Q^*(A)$ and $Q^{**}(A)$, respectively. The $Q$ range at a given $A$ is the vertical distance between the two curves.

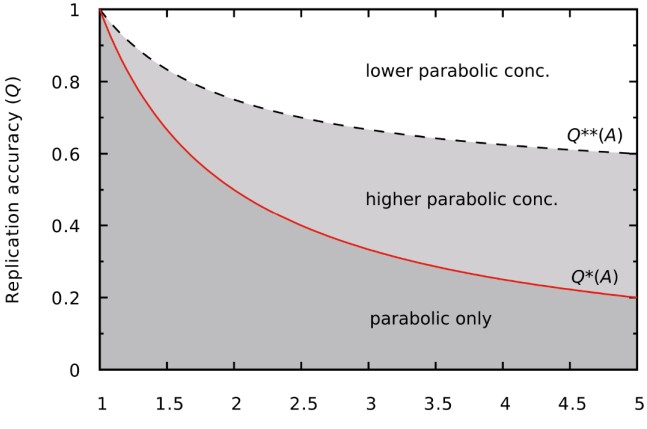

**Appendix 1—figure 1.** Different outcomes of the dynamics as a function of $Q$ and $A$. In the darker gray region (low copying fidelity, high reproductive superiority) the master survives only in the parabolic regime. In the lighter gray region (medium copying fidelity range) the master has a higher equilibrium concentration in the parabolic regime than in the exponential regime. The continuous red line is the error threshold for the exponential system ($Q^*$), the dashed black line divides the higher and lower parabolic concentration regions of the parameter space ($Q^{**}$). $a = 1$ was used.

## With back mutation

The previous model can be extended by back mutation, $B$ is the probability that a replication of a mutant results in a master. The corresponding dynamics can be read as:

$$\dot{x} = AQx + aBy - x\left(Ax + ay\right) \tag{19A}$$

$$\dot{y} = A\left(1 - Q\right)x + a\left(1 - B\right)y - y\left(Ax + ay\right) \tag{19B}$$

while its parabolic counterpart has the following form:

$$\dot{x} = AQ\sqrt{x} + aB\sqrt{y} - x\left(A\sqrt{x} + a\sqrt{y}\right) \tag{20A}$$

$$\dot{y} = A\left(1 - Q\right)\sqrt{x} + a\left(1 - B\right)\sqrt{y} - y\left(A\sqrt{x} + a\sqrt{y}\right). \tag{20B}$$

The non-trivial equilibrium master concentration of the exponential dynamics of **Equation 19A** and **Equation 19B** is:

$$\hat{x}_{eB} = \frac{AQ - a\left(B + 1\right) + \sqrt{\left[AQ - a\left(B + 1\right)\right]^2 + 4a\left(A - a\right)B}}{2\left(A - a\right)}. \tag{21}$$

Due to back mutation, the error threshold disappears from the exponential system (as a simple consequence of the Perron–Forbenius theorem), if $B > 0$ the master concentration is always positive. **Appendix 1—figure 2** shows $\hat{x}_{eB}$ as the function of $Q$ and $B$.

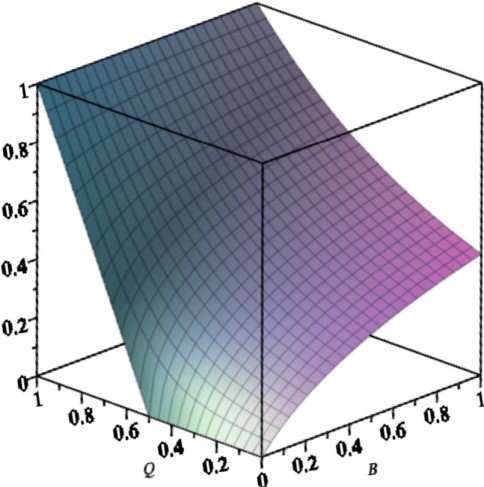

**Appendix 1—figure 2.** Equilibrium concentration of the master in the exponential system with back mutation ($\hat{x}_{eB}$) as a function of $Q$ and $B$. Parameters are: $A = 2$, $a = 1$.

As the analytical tractability of the equilibrium solution of *Equation 20A* and *Equation 20B* is limited, we carried out numerical simulations to analyze the concentrations of masters in both regimes (*Appendix 1—figure 3*). The equation of the delimiter line can be computed analytically by solving the $x_{eB} = x_{pB}$ equation for $B$ with computer algebra ($x_{pB}$ is the relevant fixed point of *Equation 20A* and *Equation 20B*). The result is:

$$B_1\left(Q\right) = Q \tag{22A}$$

$$B_2\left(Q\right) = -AQ + \frac{1}{2}\left(A + 1\right) \tag{22B}$$

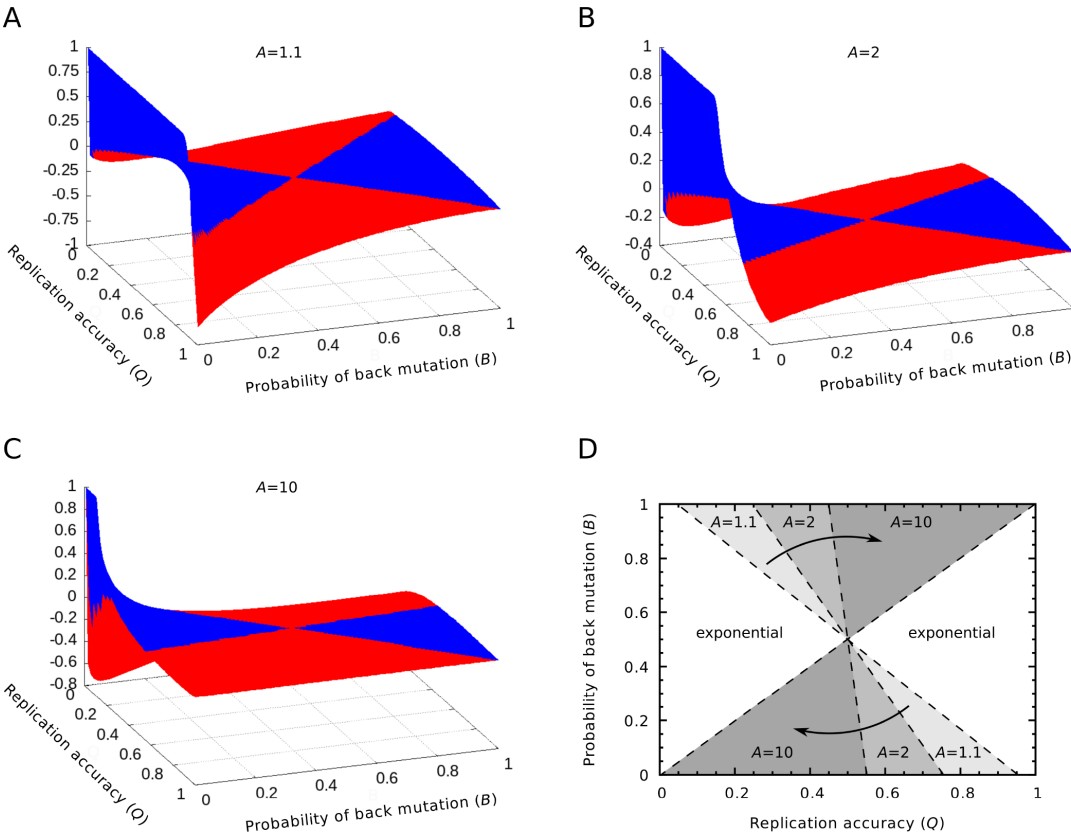

**Appendix 1—figure 3.** Relative differences between the equilibrium concentrations of the master sequences in the exponential and parabolic regimes. (**A–C**) The relative differences between the two regimes $\left(\frac{\hat{x}_{p} - \hat{x}_{e}}{\hat{x}_{p}}\right)$ are shown as the function of the replication accuracy ($Q$) and the probability of back mutation ($B$) at $A = 1.1,\ 2,\ 10$, respectively. Blue color indicates higher, red indicates lower parabolic concentrations. (**D**) The area of the parameter space where the parabolic regime has higher equilibrium concentration. The area shrinks as $A$ increases. If $A \approx a$, the parabolic case has a higher concentration in the half of the parameter space, while if $A \gg a$, this area is reduced to the quarter of the space ($a = 1$ was used).

