## [Editor Report · eLife assessment]

This study provides a **valuable** theoretical exploration of non-enzymatic sustained replication of RNA systems, in the parabolic growth regime of the evolution of putative primordial replicators. It provides **convincing** evidence that parabolic growth mitigates the error threshold catastrophe, thus demonstrating another way in which this regime contributes to the maintenance of genetic diversity. The findings shed light on relevant evolutionary regimes of primordial replicators, with potential applicability to our understanding of the origin of life.

---

## [Referee Report · Reviewer #1 (Public review)]

Summary:

Szathmary and colleagues explore the parabolic growth regime of replicator evolution. Parabolic growth occurs when nucleic acid strain separation is the rate limiting step of the replication process which would have been the case for non-enzymatic replication of short oligonucleotide that could precede the emergence of ribozyme polymerases and helicases. The key result is that parabolic replication is conducive to the maintenance of genetic diversity, that is, coexistence of numerous master sequences (the Gause principle does not apply). Another important finding is that there is no error threshold for parabolic replication except for the extreme case of zero fidelity.

Strengths:

I find both the analytic and the numerical results to be quite convincing and well described. The results of this work are potentially important because they reveal aspects of a realistic evolutionary scenario for the origin of replicators.

Weaknesses:

There are no obvious technical weaknesses. It can be argued that the results represent an incremental advance because many aspects of parabolic replication have been explored previously (the relevant publications are properly cited). Obviously, the work is purely theoretical, experimental study of parabolic replication is due. In the opinion of this reviewer, though, these are understandable limitations that do not actually detract from the value of this work.

---

## [Referee Report · Reviewer #2 (Public review)]

Summary:

A dominant hypothesis concerning the origin of life is that, before the appearance of the first enzymes, RNA replicated non-enzymatically by templating. However, this replication was probably not very efficient, due to the propensity of single strands to bind to each other, thus inhibiting template replication. This phenomenon, known as product inhibition, has been shown to lead to parabolic growth instead of exponential growth. Previous works have shown that this situation limits competition between alternative replicators and therefore promotes RNA population diversity. The present work examines this scenario in an agent-based model of RNA replication, taking into account finite population size, mutations and differences in GC content. The main results are (1) confirmation that parabolic growth promotes diversity, but that when the population size is small enough, sequences least efficient at replicating may nevertheless go extinct; (2) the observation that fitness is not only controlled by the replicability of sequences, but also by their GC content ; (3) the observation that parabolic growth attenuates the impact of mutations and, in particular, that the error threshold to which exponentially growing sequences are subject can be exceeded, enabling sequence identity to be maintained at higher mutation rates.

Strengths:

The analyses are sound and the observations intriguing. Indeed, while it has been noted previously that parabolic growth promotes coexistence, this is the first work to show that it can also mitigate the error threshold catastrophe, which is often presented as a major obstacle to our understanding of the origin of life.

Weaknesses:

A general weakness, which can however be seen as inherent in an agent-based model that aims to be more realistic than earlier, more phenomenological models, is the proliferation of parameters. The choice and values of these parameters are generally justified and, in many cases, several values are tested to assess the robustness of the results, but it can be difficult for the reader to identify the modeling choices that are truly critical from those that are less so.

---

## [Author Response]

The following is the authors’ response to the original reviews.

General remarks for the Editor and the Reviewers

We would like to thank the Editor and the Reviewers for their feedback. Below we address their comments and present our point-by-point responses as well as the related changes in the manuscript.

In addition to these changes, in a few cases we have found it necessary to move some texts and provide some additional explanations within the manuscript. We emphasize that these amendments have been made for only technical reasons, and do not alter the results and conclusions of the paper, but may help to render the text more coherent and understandable to readers with little knowledge of the subject.

These minor corrections are:

• We extended the Introduction section by a sentence (lines 40-42) that is intended to fit the proposed template directed, non-enzymatic replication mechanism into a more general prebiotic evolutionary context, thus emphasizing its biological relevance. This sentence includes an additional reference (Rosenberger et al., 2021).

• Two very methodologically oriented and repeated descriptions of random sequence generation have been moved to the Methods section (lines 178-185) from the Results section (lines 336-339 and lines 351-354).

• We complemented the Data availability statement with licensing information (lines 684-685).

• Further minor changes (also indicated by red texts) have been implemented to remedy logical and grammatical glitches.

**Public Reviews:**

**Reviewer #1 (Public Review):**
Summary:Szathmary and colleagues explore the parabolic growth regime of replicator evolution. Parabolic growth occurs when nucleic acid strain separation is the rate-limiting step of the replication process which would have been the case for non-enzymatic replication of short oligonucleotide that could precede the emergence of ribozyme polymerases and helicases. The key result is that parabolic replication is conducive to the maintenance of genetic diversity, that is, the coexistence of numerous master sequences (the Gause principle does not apply). Another important finding is that there is no error threshold for parabolic replication except for the extreme case of zero fidelity.Strengths:I find both the analytic and the numerical results to be quite convincing and well-described. The results of this work are potentially important because they reveal aspects of a realistic evolutionary scenario for the origin of replicators.Weaknesses:There are no obvious technical weaknesses. It can be argued that the results represent an incremental advance because many aspects of parabolic replication have been explored previously (the relevant publications are properly cited). Obviously, the work is purely theoretical, experimental study of parabolic replication is due. In the opinion of this reviewer, though, these are understandable limitations that do not actually detract from the value of this work.

We are grateful that this Reviewer appreciates our work. We completely agree that the ultimate validation must come from experiments. It is important to stress that in this field theory often preceded experimental work by decades, and the former often guided the latter. We hope that for the topic of the present paper experiments will follow considerably faster.

**Reviewer #2 (Public Review):**
Summary:A dominant hypothesis concerning the origin of life is that, before the appearance of the first enzymes, RNA replicated non-enzymatically by templating. However, this replication was probably not very efficient, due to the propensity of single strands to bind to each other, thus inhibiting template replication. This phenomenon, known as product inhibition, has been shown to lead to parabolic growth instead of exponential growth. Previous works have shown that this situation limits competition between alternative replicators and therefore promotes RNA population diversity. The present work examines this scenario in a model of RNA replication, taking into account finite population size, mutations, and differences in GC content. The main results are (1) confirmation that parabolic growth promotes diversity, but that when the population size is small enough, sequences least efficient at replicating may nevertheless go extinct; (2) the observation that fitness is not only controlled by the replicability of sequences, but also by their GC content; (3) the observation that parabolic growth attenuates the impact of mutations and, in particular, that the error threshold to which exponentially growing sequences are subject can be exceeded, enabling sequence identity to be maintained at higher mutation rates.Strengths:The analyses are sound and the observations are intriguing. Indeed, it has been noted previously that parabolic growth promotes coexistence, its role in mitigating the error threshold catastrophe - which is often presented as a major obstacle to our understanding of the origin of life - had not been examined before.Weaknesses:Although all the conclusions are interesting, most are not very surprising for people familiar with the literature. As the authors point out, parabolic growth is well known to promote diversity (SzathmaryGladkih 89) and it has also been noted previously that a form of Darwinian selection can be found at small population sizes (Davis 2000).Given that under parabolic growth, no sequence is ever excluded for infinite populations, it is also not surprising to find that mutations have a less dramatic exclusionary impact.

In the two articles cited (Szathmary-Gladkih 1989 and Davis 2000) the subexponentiality of the system was implemented in a mechanistic way, by introducing the exponent 0<p<1. Although the behaviour of these models is more or less consistent with experimental findings (von Kiedrowski, 1986; Zielinski and Orgel, 1987), the divergence of per capita growth rates (x˙/x) at very low concentrations–which guarantees the ability to maintain unlimited diversity in the case of infinite population sizes–makes this formal approach partly unrealistic.

To avoid the possible artefacts of this mechanistic approach, and as there are no previous studies analysing the diversity maintaining ability of finite populations of parabolic replicators in an individual-based model context, we implemented a simplified template replication mechanism leading to parabolic growth and analysed the dynamics in an individual-based stochastic model context. The key point of our investigation is that considerable diversity can be maintained in the system even when the population size is quite small.

Regarding the Reviewer’s comment on selection: Darwinian selection can only occur in a simple subexponential dynamics if the ratio of replicabilities diverges, cf. Eq. (8) and the preceding paragraph in Davis, 2000.

Our results also show (Figs. 4B and 4C) that high mutation rates and the error threshold problem can still be considered as a major limiting factor for parabolically replicating systems in terms of their diversity-maintaining ability. In the light of the above, potential mechanisms to relax the error threshold in such systems, one of which is demonstrated in the present study, seem to be important steps to account for the sequence diversification and increase in molecular complexity during the early evolution of RNA replicators.

A general weakness is the presentation of models and parameters, whose choices often appear arbitrary. Modeling choices that would deserve to be further discussed include the association of the monomers with the strands and the ensuing polymerization, which are combined into a single association/polymerization reaction (see also below), or the choice to restrict to oligomers of length L = 10. Other models, similar to the one employed here, have been proposed that do not make these assumptions, e.g. Rosenberger et al. Self-Assembly of Informational Polymers by Templated Ligation, PRX 2021. To understand how such assumptions affect the results, it would be helpful to present the model from the perspective of existing models.

The assumption of one-step polymerization reactions that we used here is a common technique for modelling template replication of sequence-represented replicators [see, e.g., Fontana and Schuster, 1998 (10.1126/science.280.5368.1451), Könnyű et al., 2008 (10.1186/1471-2148-8267), Vig-Milkovics et al, 2019 (10.1016/j.jtbi.2018.11.020) or Szilágyi et al., 2020 (10.1371/journal.pgen.1009155)]. This is because assuming base-to-base polymerisation of the copy would lead to a very large number of different types of intermediates, which a Gillespietype stochastic simulation algorithm could not handle in reasonable computation times, even if the sequences were relatively short. For comparison, in our model, where polymerization is one-step, the characteristic time of a simulation for L=10, N=105 and δ=0.01 was 552 hours.

Note that in Rosenberg et al. (PRX 2021), in contrast to a pioneering work [Fernando et al, 2007 (10.1007/s00239-006-0218-4)], sequences of replicators are not represented, which makes this approach completely inapplicable to our case, in which sequence defines the fitness. In sum, we suggest that this valid criticism points to possible future work.

The values of the (many) parameters, often very specific, also very often lack justifications. For example, why is the "predefined error factor" ε = 0.2 and not lower or higher? How would that affect the results?

A general remark. For the more important parameters (N,δ,pmut,f,cin,cδds,cout,cr,cd), several values were used to test the behaviour of the model (see Table 1), but due to the considerable number of parameters, it is impossible to examine all possible combinations. ca=1 fixes the timescale, L is set to 10 to obtain reasonable running times (see above).

εcharacterizes how replicability decreases as the number of mutations increases. In the manuscript we used the following default vector: ε=(0.05,0.2,1) in which the third element corresponds to the mutation-free sequence, so it must to be 1. The first element determines the baseline replicability (see Methods), which we preferred not to change because it would fundamentally alter the ratio of replication propensities to association and dissociation propensities (as the substantial amount of complementary sequences of the master sequences are of baseline replicability) and thus would alter the reaction kinetics to an extent that it is not comparable with the original results. Therefore, only the second element can be adjusted. Accordingly, we have analysed the behaviour of the model in the cases of a steeper and a more gradual loss of replicability using the following two vectors, respectively: ε′=(0.05,0.05,1) and ε′′=(0.05,0.5,1). The choice of ε′ is chemically more plausible, since for very short oligomers the loss of chemical activity and replicability as a function of the number of mutations can be very sharp. We performed a series of simulations with all possible combinations of δ=0.001,0.005,0.1 and N=103,104,105 for ε′ and ε′′ in the constant population and chemostat model context (36 different runs). For other parameters, we took the default values, see Table 1. These values also correspond to the parameters we used in Figures 2 and 6. The results show that the steeper loss of replicability (ε′) slightly increases the diversity maintaining ability of the system, whereas the more gradual loss of replicability (ε′′) moderately decreases the diversity-maintaining ability of the system, and that these shifts are more pronounced in the constant population size model (Author response image 1) than in the chemostat model (Author response image 2). Altogether, these results confirm that the qualitative outcome of the model is robust in a wide range of loss of replicability (ε vector) values.

**Author response image 1. sa3fig1:** Replicator coexistence in the constant population model with different loss of replicability (𝜀 vector) values. Within a given combination of 𝛿 and 𝑁 parameter values, the upper panel corresponds to the steeper loss of replicability (𝜀!), the middle panel to the default 𝜀 vector (Figure 2A), and the bottom panel to the more gradual loss of replicability vector (𝜀!!). Within each 𝛿; 𝑁 parameter combination, the same master sequence set was used with the three different 𝜀 vectors for comparability.

**Author response image 2. sa3fig2:** Replicator coexistence in the chemostat model with different loss of replicability (𝜀 vector) values. Within a given combination of 𝛿 and 𝑁 parameter values, the upper panel corresponds to the steeper loss of replicability (𝜀!), the middle panel to the default 𝜀 vector (Figure 6A), and the bottom panel to the more gradual loss of replicability vector (𝜀!!). Within each 𝛿; 𝑁 parameter combination, the same master sequence set was used with the three different 𝜀 vectors for comparability.

Similarly, in equation (11), where does the factor 0.8 come from?

This factor scales the decay rate of duplex sequences (cδds) as the function of the binding energy (Eb). The value of 0.8 is an arbitrary choice, the value should be in the interval (0,1) and is only relevant in the chemostat model. It is expected to have a similar effect on the dynamics as the duplex decay factor parameter f, which we have investigated in a wide range of different values (cf. Table 1, Fig. 6), although f is independent of the binding energy (Eb): increasing/decreasing the 0.8 factor is expected to decrease/increase the average total population size. We have investigated the diversity maintaining ability of the system at smaller (0.6) and larger (0.9) parameter values at different population sizes (N ≈ 10^3^, 10^4^ and 10^5^) and at different replicability distances (δ = 0.001, 0.005 and 0.01) as shown in Fig. 6. We have found that the number of coexisting master types changes very little in response to changes in this factor. Only two shifts could be detected (underlined): factor 0.9 combined with N≈104 and δ=0.001 caused the number of surviving master types to decrease by one, while factor 0.9 combined with N≈103 and δ=0.01 caused the number of surviving master types to increase by one (Author response table 1). Factor 0.6 produced the same number of surviving types as the default (Author response table 1). In summary, the model shows marked robustness to changes in the values of this parameter.

**Author response table 1. sa3table1:** Number of coexisting master types in the chemostat model with different binding energy dependent duplex decay rates. Within each 𝛿; 𝑁 parameter combination, the same master sequence set was used with the three different factor values: 0.6, 0.8 (the original) and 0.9 for comparability.

	*N* ≈ 10^3^	*N* ≈ 10^4^	*N* ≈ 10^5^	
Cδds=f⋅Cδss⋅0.6Eb	4	8	9	δ=0.001
Cδds=f⋅Cδss⋅0.8Eb (default)	4	8	9
Cδds=f⋅Cδss⋅0.9Eb	4	7	9
Cδds=f⋅Cδss⋅0.6Eb	3	6	7	δ=0.005
Cδds=f⋅Cδss⋅0.8Eb (default)	3	6	7
Cδds=f⋅Cδss⋅0.9Eb	3	6	7
Cδds=f⋅Cδss⋅0.6Eb	2	5	6	δ=0.01
Cδds=f⋅Cδss⋅0.8Eb (default)	2	5	6
Cδds=f⋅Cδss⋅0.9Eb	3	5	6

Why is the kinetic constant for duplex decay reaction 1.15e10−8?

Note that this value is the minimum of the duplex decay rate, Table 1 correctly shows the interval of this kinetic constant as: [1.15 ⋅ 10^-8^, 6.4 ⋅ 10^-5^]. Both values are derived from the basic parameters of the system (f, cδss and Eb) and can be computed according to Eq. (11). The minimum: cδss=f⋅cδss⋅0.8Eb=0.01⋅10−4⋅0.8−20=1.15⋅10−8 as the parameter set corresponding to this value is: Eb=20. The maximum: cδss=f⋅cδSS⋅0.8Eb=1⋅10−4⋅0.82=6.4⋅10−5 with Eb=2.

Are those values related to experiments, or are they chosen because specific behaviors can happen only then?

See above.

The choice of the model and parameters potentially impact the two main results, the attenuation of the error threshold and the role of GC content:Regarding the error threshold, it is also noted (lines 379-385) that it disappears when back mutations are taken into account. This suggests that overcoming the error threshold might not be as difficult as suggested, and can be achieved in several ways, which calls into question the importance of the particular role of parabolic growth. Besides, when the concentration of replicators is low, product inhibition may be negligible, such that a "parabolic replicator" is effectively growing exponentially and an error catastrophe may occur. Do the authors think that this consideration could affect their conclusion? Can simulations be performed?

The assumption of back mutation only provides a theoretical solution to the error threshold problem: back mutation guarantees a positive (non-zero) concentration of a master type, but, since the probability of back mutation is generally very low, this equilibrium concentration may be extremely low, or negligible for typical system sizes. Consequently, back mutation alone does not solve the problem of the error catastrophe: in our system back mutation is present (the probability that a sequence with k errors mutates back to a master sequence is μk(1−μ)L−k, and the diversity-maintaining ability is limited). The effect of back mutation decreases exponentially with increasing sequence length.

Regarding the role of the GC content, GC-rich oligomers are found to perform the worst but no rationale is provided.

For GC-rich oligonucleotides the dissociation probability of a template-copy complex is relatively low (cf. Eqs. (9, 10)), thus they have a relatively low number of offspring, cf. lines 557-561: “a relatively high dissociation probability and the consequential higher propensity of being in a simple stranded form provides an advantage for sequences with relatively low GC content in terms of their replication affinity, that is, the expected number of offspring in case of such variants will be relatively high.”. Note that the simulation results shown in Fig. 3A, demonstrate the realization of this effect with prepared sequences (along a GC content gradient).

One may assume that it happens because GC-rich sequences are comparatively longer to release the product. However, it is also conceivable that higher GC content may help in the polymerization of the monomers as the monomers attach longer on the template (as described in Eq. (9)). This is an instance where the choice to pull into a single step the association and polymerization reactions are pulled into a single step independent of GC content may be critical.It would be important to show that the result arises from the actual physics and not from this modeling choice.Some more specific points that would deserve to be addressed:Line 53: it is said that p "reflects how easily the template-reaction product complex dissociates". This statement is not correct. A reaction order p<1 reflects product inhibition, the propensity of templates to bind to each other, not slow product release. Product release can be limiting, yet a reaction order of 1 can be achieved if substrate concentrations are sufficiently high relative to oligomer concentrations (von Kiedrowski et al., 1991).

We think the key reference is Von Kiedrowski (1993) in this case. Other things being equal, his Table 1 on p. 134 shows that a sufficient increase in 𝐾4, i.e., the stability of the duplex (template and copy) (association rate divided by dissociation rate) throws the system into the parabolic regime. This is what we had in mind. In order to clarify this, we modified the quoted sentence thus: “In this kinetics, the growth order is equal or close to 0.5 (i.e., the dynamics is sub-exponential) because increased stability of the template-copy complex (rate of association divided by dissociation) promotes parabolic growth (von Kiedrowski et al., 1991; von Kiedrowski & Szathmáry, 2001).”

Population size is a key parameter, and a comparison is made between small (10^3) and large (10^5) populations, but without explaining what determines the scale (small/large relative to what?).

The “small” value (103) corresponds to the smallest meaningful population size, significantly smaller population sizes (e.g. 102) cannot maintain the 10 master types (or any subset of them) and are chemically unrealistic. The “large value” (105) is the largest population size for which simulation times are still acceptable, in the case of 106 the runtimes are in the order of months.

In the same vein, we might expect size not to be the only important parameter, but also concentration.

With constant volume population size and concentration are strictly coupled.

Lines 543-546: if understanding correctly, the quantitative result is that the error threshold rises from 0.1 in the exponential case to 0.196 in the parabolic. Are the authors suggesting that a factor of 2 is a significant difference?

In this paragraph we compared the empirical error threshold of our system (which is close to pmut=0.15) with the error threshold of the well-known single peak fitness landscape (which can be approximated by 1L=0.1,(L=10)) as a reference case. To make the message even clearer we have extended the last sentence (lines 596-597) as follows: “but note that applying this approach to our system is a serious oversimplification”. The 0.196 is simply the probability of error-free replication of a sequence when pmut=0.15:Q=(1−pmut)L=0.196, but we have removed this sentence (“corresponding to the Q=(1−0.15)10=0.196 replication accuracy of a master sequence”) from the manuscript as it seems to be confusing.

Figure 3C: this figure shows no statistically significant effect?

Thank you for pointing out this. We statistically tested the hypothesis that the GC content between the survived and the extinct master subsets are different. This analysis revealed that the differences between these two groups are statistically significant, which we now included in the manuscript at lines 380-390: “A direct investigation of whether the sequence composition of the master types is associated with their survival outcome was conducted using the data from the constant population model simulation results (Figure 2). In these data, the average GC content was measured to be lower in the surviving master subpopulations than in the extinct subpopulations (Figure 3C). To determine whether this difference was statistically significant, nonparametric, two-sample Wilcoxon rank-sum tests (Hollander & Wolfe, 1999) were performed on the GC content of the extinct-surviving master subsets. The GC content was significantly different between these two groups in all nine investigated parameter combinations of population size (N) and replicability distance (δ) at p<0.05 level, indicating a selective advantage for a lower GC content in the constant population model context. The exact p values obtained from this analysis are shown in Figure 3C.”

line 542: "phase transition-like species extension (Figure 4B)": such a clear threshold is not apparent.

Thank you for pointing out the incorrect phrasing. As there is no clear threshold in the number of coexisting types as a function of the mutation rate, we removed the “phase transition-like” expression: “However, when finite population sizes and stochastic effects are taken into account, at the largest investigated per-base mutation rate (𝑝mut = 0.15), the summed relative steady-state master frequencies approach zero (Figure 4C) with accelerating species extinction (Figure 4B), indicating that this value is close to the system׳s empirical error threshold.” (lines 589-594).

**Recommendations for the authors:**

**Reviewer #1 (Recommendations For The Authors):**
On the whole, the work is well done and presented, there are no major recommendations. It seems a good idea to cite and briefly discuss this recent paper: https://pubmed.ncbi.nlm.nih.gov/36996101/ which develops a symbiotic scenario of the coevolution of primordial replicators and reproducers that appears to be fully compatible with the results of the current work.

Thank you for bringing this article to our attention. We have inserted the following sentence at lines 621-624: “The demonstrated diversity-maintaining mechanism of finite parabolic populations can be used as a plug-in model to investigate the coevolution of naked and encapsulated molecular replicators (e.g., Babajanyan et al., 2023).”

The manuscript is well written, but there are some minor glitches that merit attention. For example:l. 5 "carriers presents a problem, because product formation and mutual hybridization" - "mutual" is superfluous here, deletel. 13 "amplification. In addition, sequence effects (GC content) and the strength of resource" - hardly "effects" - should be 'features' or 'properties'l. 41 "If enzyme-free replication of oligomer modules with a high degree of sequence" - "modules" here is only confusing - simply, "oligomers"l. 44 "under ecological competition conditions with which distinct replicator types with different" - delete "with" etc, there are many such minor glitches that are best corrected.

Thank you for pointing out, we have corrected! Other drafting errors, glitches, superfluous sentences have also been corrected.

**Reviewer #2 (Recommendations For The Authors):**

None

**Editor (Recommendations For The Authors):**
In the manuscript, it appears that coexistence is assessed at a given point in time, while figures seem to show that it remains time-dependent. It would be great if the authors could clarify this and/or discuss this.

We appreciate you bringing this to our attention, as we have indeed missed to elaborate on this important point. The steady state characteristic of the coexistence is assessed in our model in the following way: the relative frequency of each master sequence is tested for the condition of ≥ 100- (cut-off relative frequency for survival) in every 2,000th replication step in the interval between 10,000 replication steps before termination and actual termination (10 = replication steps). If the above condition is true more than once, we consider the master type in question as survived (we have included this explanation in the Methods section: lines 258-268). Although this relatively narrow time interval can still be regarded as a snapshot of the state of the system, according to our numerical experiences, the resulting measure is a reliable quantitative indicator of the apparent stability of species coexistence in the parabolic dynamics.